# Strengthening phage resistance of *Streptococcus thermophilus* by leveraging complementary defense systems

Audrey Leprince [1] ✉, Justine Lefrançois[1], Anne M. Millen[2], Damian Magill[3], Philippe Horvath [3], Dennis A. Romero[2], Geneviève M. Rousseau [1] & Sylvain Moineau [1] ✉

CRISPR-Cas and restriction-modification systems represent the core defense arsenal in *Streptococcus thermophilus*, but their effectiveness is compromised by phages encoding anti-CRISPR proteins (ACRs) and other counter-defense strategies. Here, we explore the defensome of 263 *S. thermophilus* strains to uncover other anti-phage systems. The defense landscape of *S. thermophilus* is enriched by 21 accessory defense systems, 13 of which have never been investigated in this species. Experimental validation of 17 systems with 14 phages reveals a range of anti-phage activities, highlighting both broad and narrow specificities across the five viral genera infecting *S. thermophilus*. Synergies are observed when combining CRISPR immunity with accessory systems. We also assess the fitness cost associated with the chromosomal integration of these systems in their native context and find no impact under laboratory or industrial conditions. These findings underscore the potential of these accessory defense systems to enhance the resistance of *S. thermophilus*, particularly against ACR-encoding phages.

*Streptococcus thermophilus* is a lactic acid bacterium (LAB) widely used to produce fermented dairy products such as yogurt and certain cheeses. In the dairy industry, large-scale milk fermentation processes are meticulously controlled to ensure the quality and consistency of the final product. However, phages present a significant risk for these added bacterial cultures as they can reduce bacterial growth and milk acidification rates, thereby disrupting the fermentation process[1]. These delays or halts in fermentation can result in poor-quality products and economic losses. Therefore, controlling virulent phages in this environment requires constant monitoring, explaining why phages that infect LAB have received significant research attention[2]. Because phages are impossible to eliminate from manufacturing facilities due to constant entry through milk supply, the dairy industry relies on a variety of control strategies to mitigate phage contamination and dissemination[3]. One crucial approach is using industrial strains with increased resistance against circulating phages[2]. To date, five genera are known to infect *S. thermophilus*, with phages belonging to the *Moineauvirus* (formerly *cos* group) and the *Brussowvirus* (*pac* group) genera being the most predominant, isolated in 69% and 29% of cases, respectively[2]. Members of the three other genera, *Vansinderenvirus* (5093 group), *Piorkowskivirus* (987 group), and P738 are more rarely isolated.

Bacteria have evolved multiple defense systems to counter phage infection at different stages of the viral replication cycle[4]. In *S. thermophilus* strains, Clustered Regularly Interspaced Short Palindromic Repeats and associated protein (CRISPR-Cas) systems are prevalent and have been well characterized[5,6]. By challenging sensitive strains with problematic phages, strains that acquire additional spacers and

[1]Département de biochimie, de microbiologie et de bio-informatique, Faculté des sciences et de génie, Université Laval, Québec City, QC, Canada. Institut de Biologie Intégrative et des Systèmes (IBIS), Pavillon Charles-Eugène-Marchand, Université Laval, Québec City, QC, Canada. [2]IFF Health and Biosciences, 3329 Agriculture Dr, Madison, WI, USA. [3]IFF Health and Biosciences, CS 10010, Dangé-Saint-Romain, France. ✉e-mail: audrey.leprince.1@ulaval.ca; sylvain.moineau@bcm.ulaval.ca

exhibit increased resistance can be selected and used in starter culture blends or included in strain rotation strategies[7]. In addition, this relatively minor genomic modification does not affect the strain's other technological properties. As this adaptive immunity system can naturally improve the phage resistance profile, the derived industrial cultures do not require regulatory approval. Thus, CRISPR-Cas systems have become fundamental in generating bacteriophage-insensitive mutants (BIMs) that are now used worldwide[7,8].

However, due to the constant evolutionary race with their host, phages have developed counter-defense mechanisms to bypass CRISPR-Cas immunity[9]. For instance, CRISPR escape mutants (CEMs) have acquired mutations in their protospacer or protospacer adjacent motif (PAM) to prevent matching with their host spacer, thereby abolishing interference[10,11]. In addition, several *S. thermophilus* phages encode one or multiple anti-CRISPR (ACR) proteins that neutralize Cas9, thereby preventing DNA binding or nuclease activity[12–14]. Therefore, it is crucial to explore alternative strategies for generating phage-resistant strains to safeguard CRISPR-Cas defense.

The diversity of anti-phage defense systems has been extensively explored in recent years, uncovering over 250 distinct systems with diverse mechanisms from various bacterial species[15,16]. In *S. thermophilus*, although research has primarily focused on CRISPR-Cas systems, early studies also characterized several restriction-modification (RM) systems[17–19] as well as a few prophage-encoded lipoproteins, the latter protecting against related phages through superinfection exclusion[20,21]. More recently, Kelleher et al.[22], while investigating the methylome of 27 industrial *S. thermophilus* strains, highlighted the significant role RM systems play in defense in this species. They also identified nine additional defense systems in these strains, with seven of them (AbiD, AbiE, Gao19, Gabija, Hachiman, Kiwa, and SoFic) conferring low-to-moderate resistance against four streptococcal phages[22]. Through the analysis of other *S. thermophilus* genomes, they also identified, but not tested, five other defense systems (AbiH, Borvo, PD-Lambda-1, PrrC, and RloC)[22].

In this study, we analyze known defense systems in publicly available genomes of *S. thermophilus* and counter-defense strategies in streptococcal phages. We identify 28 defense systems, showing that CRISPR-Cas and RM systems function as core lines of defense. However, we also show that phages have developed multiple counter-defense strategies to bypass these systems, highlighting the need for a deeper investigation of other defense mechanisms. Among the 21 other defense systems identified, we show the effectiveness of 17 of them against dairy phages. Importantly, by combining some of the most effective defense systems with CRISPR-Cas, we enhance the overall resistance levels, especially against phages that encode ACRs.

## Results

### *S. thermophilus* defensome expands beyond CRISPR-Cas and RM systems

Defense systems were predicted in publicly available genomes of *S. thermophilus* (N = 263) using DefenseFinder[23] and PADLOC[24] (Fig. 1A and Supplementary Data 1 and 2). While both tools generally provided similar predictions, we noticed that each missed some defense systems (Supplementary Data 2). For example, type IV RM systems were only detected by PADLOC, while AbiH was only identified by DefenseFinder, despite both bioinformatic tools containing models for these systems. By combining their outputs, we achieved a more comprehensive analysis across our dataset, which includes incomplete genomes, enabling the detection of less common defense systems (Fig. 1A). This approach revealed 28 distinct defense systems, expanding the known diversity of the *S. thermophilus* defensome[22]. Among these, we identified three CRISPR-Cas types (II-A, III-A, and I-E), four RM types (I-IV), and 21 additional defense systems, 13 of which were not previously tested in Kelleher et al.[22]. The number of defense systems per strain ranged from 3 to 13, with an average of 7.5 systems per strain (Fig. 1B), which is

similar to the numbers reported by Kelleher et al.[22]. Plasmids and prophages are rare in *S. thermophilus*, detected in only 12.9% (34/263) and 6.8% (18/263) of the strains, respectively (Supplementary Data 1). Consequently, except for four RM systems and one AbiD homolog, all systems were chromosomally encoded, and none were found within prophages (Supplementary Data 2). Nonetheless, a comparison of defense system distribution across strain phylogeny revealed that they are not only shared among closely related strains but also between strains from different clades, suggesting that horizontal gene transfer mechanisms, notably natural competence, which is common in *S. thermophilus*, play a role in their dissemination (Fig. 1C and Supplementary Fig. 1).

### CRISPR-Cas and RM systems are extensively found in *S. thermophilus*

CRISPR-Cas systems have been widely investigated in *S. thermophilus*, due to their prevalence and ability to be easily manipulated for industrial purposes. Previous studies identified four distinct CRISPR loci (CR1-4)[25,26]. Among these, two well-known type II-A CRISPR-Cas systems associated with the CR1 and CR3 loci were highly prevalent in *S. thermophilus* strains (Figs. 1A and 2A). Specifically, the CR1 locus was present in all strains, while 70% (184/263) also harbored the CR3 locus (Supplementary Data 3). These two CRISPR loci have been shown to be very effective at acquiring new spacers[27]. The CR2 locus, classified as a subtype III-A system, was found in 88% (232/263) of the strains. Finally, the CR4 locus, belonging to a type I-E system, was present in only 9% (23/263) of the strains. Although rare, in vivo spacer acquisition in the CR2 locus was recently demonstrated for one strain[28], however no such acquisition was ever observed for CR4[25]. CRISPR loci frequently co-occurred within the same genome, with most strains encoding two (31%, 81/263) or three (59%, 155/263) loci, suggesting a complementary role of these types of CRISPR-Cas systems[29] (Fig. 2A). Of note, while type II-C have been reported in studies focusing on in silico prediction of CRISPR-Cas systems in *S. thermophilus*, our analysis identified only one such instance, which turned out to be a misprediction of a type II-A system, confirming the absence of type II-C in this species.

RM systems were also prevalent in *S. thermophilus*, with at least one RM system present in 98% (257/263) of the analyzed strains, and some harboring up to nine distinct RM systems (Fig. 2B and Supplementary Fig. 2). Type I RM systems were nearly ubiquitous, consistent with the methylome analysis of 27 *S. thermophilus* strains, which attributed most methylated motifs to type I RM, with only a few linked to type II and III RM systems[22]. Type IV RM systems that cleave methylated DNA were found in 33% (88/263) of the strains (Fig. 2C). Like CRISPR-Cas systems, different RM types frequently co-occurred within the same strain (95%, 250/263), likely due to their mode of action, sequence specificities and varying susceptibilities towards phage-encoded anti-RM proteins (Fig. 2B). Multiple instances of the same RM type within a single strain were found for types I, II, and to a lesser extent, III (Fig. 2C). Some strains harbored up to three type I or six type II RM systems. Predictions of putative recognition sites for type II RM systems showed that these co-occurring systems often recognize different sequences, thereby enhancing the strain's ability to target a broader range of phages with varying restriction site frequencies in their genomes (Fig. 2D and Supplementary Fig. 2B). Similar trends were also observed for type I and III RM systems (Supplementary Fig. 2C, D).

### Streptococcal phages counter CRISPR-Cas and RM defenses

Given the widespread presence of CRISPR-Cas and RM systems in *S. thermophilus*, we investigated counter-defense strategies in the publicly available genomes of their infecting phages (N = 191).

One strategy used by phages to evade CRISPR-Cas targeting is through mutations in their protospacer sequences or PAM motifs. Consequently, protospacer–spacer dynamics represent a key aspect of

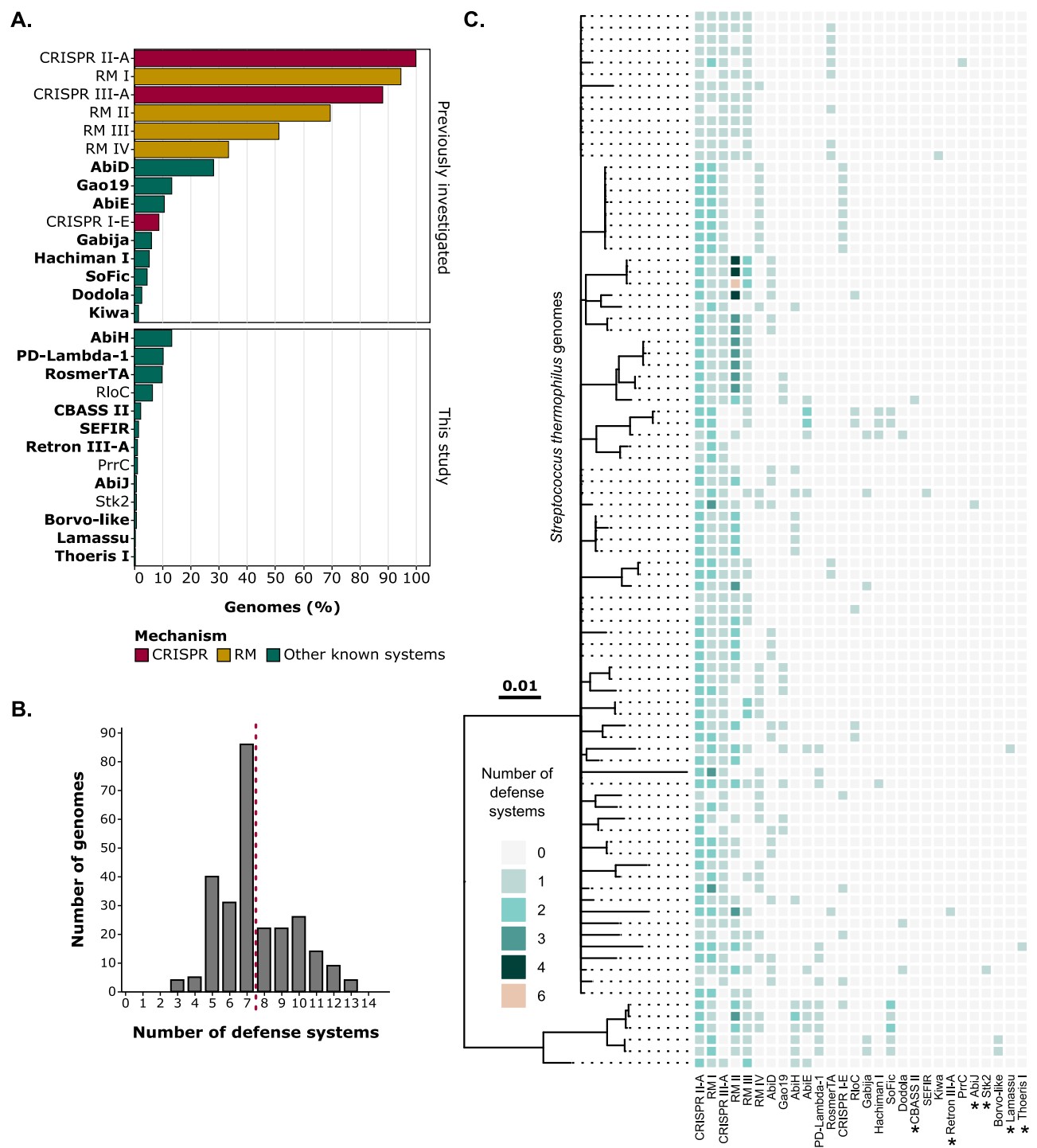

**Fig. 1 | *S. thermophilus* anti-phage defense systems extend beyond CRISPR-Cas and RM systems. A** Prevalence of known defense systems in *S. thermophilus* (*N* = 263, RefSeq as of March 2023). These systems are categorized into previously reported and newly identified in this study. The 18 systems experimentally validated in this work are highlighted in bold. The "Borvo-like" system differs from the original Borvo, being annotated as a two-gene system composed of two BovA subunits, rather than one. **B** Distribution of the number of defense systems per genome. The red dashed line indicates the average number of defense systems per *S. thermophilus* genome. **C** Phylogenetic tree of complete *S. thermophilus* genomes (*N* = 85), illustrating the distribution of identified defense systems. Six incomplete genomes were included to show systems which were not found in complete genomes (marked with an asterisk). Branch lengths represent the number of amino acid substitutions per site. Source data are provided as a Source Data file.

the co-evolutionary arms race between phages and bacteria, particularly in *S. thermophilus*, where CRISPR-Cas systems are highly prevalent (Supplementary Fig. 3). In addition to this indirect mechanism, six families of known ACRs that target type II-A systems were identified in 40% (77/191) of the streptococcal phages (Fig. 2E). Among these, AcrIIA6, which neutralizes the Cas9 nuclease associated with the CR1

locus, was especially frequent. AcrIIA6 was found in 36% (69/191) of the phage genomes, including in 42% (49/116) of the *Moineauvirus* genus, the most prevalent streptococcal viral genus in the dairy industry[2]. AcrIIA5, which inhibits both type II-A CRISPR-Cas systems (CR1 and CR3) and AcrIIA3, which inhibits only the system linked to the CR3 locus, were present in 4.2% (8/191) and 2.6% (5/191) of the phages,

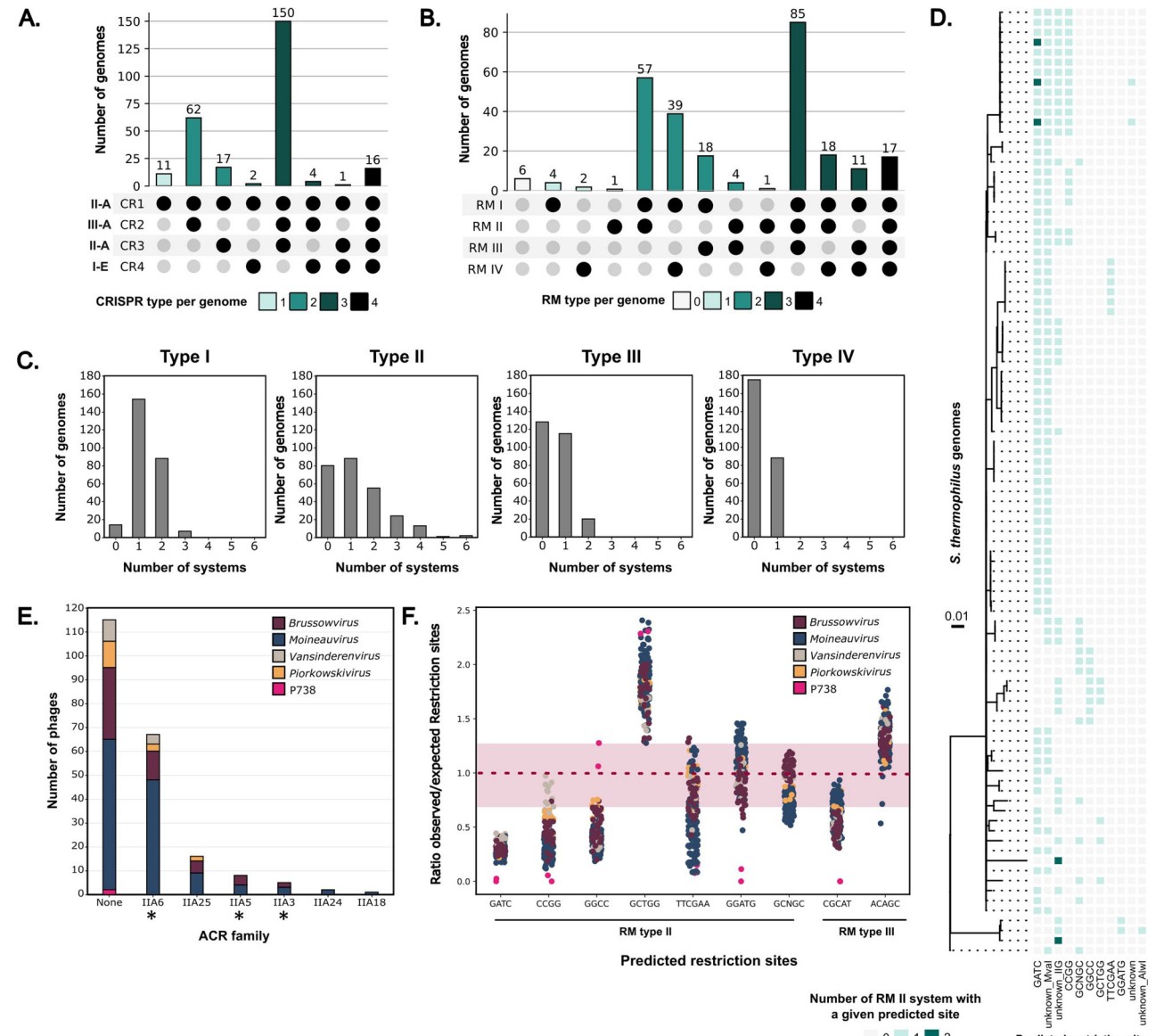

**Fig. 2 | CRISPR and RM systems are widespread in *S. thermophilus*, and phages have several counter defense mechanisms.** Co-occurrence of (**A**). CRISPR-Cas loci and (**B**). RM types within the same *S. thermophilus* genome. **C** Distribution of the number of RM systems per genome according to the type. **D** Heatmap of co-occurrence within the same genome of RM type II with different predicted restriction sites. Only strains harboring at least two RM type II systems are shown on the phylogenetic tree (*N* = 95). RM systems with no predicted restriction site were divided into different unknown categories based on the restriction enzyme annotation (MvaII or AlwI) or subtype (IIG). **E** Prevalence of predicted anti-CRISPR (ACR) proteins in phages infecting *S. thermophilus* (*N* = 191, NCBI June 2023). Experimentally validated ACRs in *S. thermophilus* are marked with a star. Color coding corresponds to phage genus. **F** Plot of the ratios of observed to expected number of restriction sites in streptococcal phages for RM type II and III restriction sites (predicted with REBASE). The observed number of sites corresponds to the count of each site within the phage genomes, while the expected number was estimated using a Markov-immediate neighbor dependence statistical model[68]. Ratios < 0.75 indicate underrepresentation and ratios > 1.25 indicate overrepresentation. Source data are provided as a Source Data file.

respectively. AcrIIA25, AcrIIA24, and AcrIIA18, which have not been characterized yet in *S. thermophilus* phages, were identified in 8.3% (16/191), 1.5% (3/191) and 0.5% (1/191) of the phage genomes analyzed, respectively. Overall, ACRs were more prevalent in phages belonging to the *Moineauvirus* (46%, 53/116) and *Brussowvirus* (35%, 16/46) genera, as compared to phages of the rarely found *Piorkowskivirus* (21%, 3/14), *Vansinderenvirus* (31%, 4/13), and P738 (none, 0/2).

No known anti-RM protein was detected in the streptococcal phages analyzed. However, DNA methyltransferases were present in 20% of them, indicating that some may evade genome restriction through DNA methylation of recognition sites (Supplementary Data 4). In addition, previous studies have shown that reducing the number of restriction sites within phage genomes increases the likelihood of evading RM[30]. An analysis of restriction site avoidance for the predicted recognition sites of type II and III RM systems in streptococcal phages revealed that certain restriction sites, such as GATC, CCGG, and GGCC, were commonly underrepresented in most phage genomes (Fig. 2F). Others, such as GGATG, were selectively excluded in some viral genera (e.g., P738-like phages). Conversely, the predicted GCTGG site showed no reduction in number across all phages. These data suggest that the reduced numbers of restriction sites seem to be a common evasion strategy against RM systems in streptococcal phages.

## Versatile modes of action in accessory defense systems

Our bioinformatic analysis uncovered 21 non-CRISPR and non-RM defense systems (Fig. 1A). AbiD was the most prevalent, occurring in

**Table 1 | Information on additional defense systems found in *S. thermophilus***

| Defense system | Sub-type | Total number | Number of homologs | Maximum number per strain | Mechanism | Abi phenotype | Ref. |
|---|---|---|---|---|---|---|---|
| **Broad activity** | | | | | | | |
| Gao19 | – | 35 | 12 | 1 | NAD+/ATP depletion, DNA degradation | Yes | 34 |
| Gabija | – | 16 | 7 | 1 | DNA degradation | Yes | 35 |
| Hachiman | I | 14 | 5 | 1 | DNA degradation | Yes | 36 |
| Dodola | – | 7 | 3 | 1 | Unknown | Unknown | 16 |
| Thoeris | I | 1 | 1 | 1 | NAD+ depletion; Signaling-based | Yes | 33 |
| **Narrow activity** | | | | | | | |
| AbiD | – | 76 | 15 | 2 | Unknown | Yes | 77 |
| AbiH | – | 36 | 12 | 2 | Unknown | Yes | 78 |
| AbiE | – | 35 | 9 | 2 | Toxin anti-toxin | Yes | 39 |
| PD-Lambda-1 | – | 27 | 10 | 1 | Unknown | Unknown | 68 |
| RosmerTA | – | 26 | 4 | 1 | Toxin anti-toxin | Yes | 16 |
| SoFic | – | 14 | 4 | 2 | Unknown | Unknown | 16 |
| CBASS | II | 6 | 1 | 1 | Signaling-based | Yes | 38 |
| Kiwa | – | 4 | 1 | 1 | DNA degradation | No | 37 |
| SEFIR | – | 4 | 2 | 1 | NAD+ depletion | Yes | 16 |
| Borvo-like | – | 2 | 1 | 1 | Unknown | Yes | 16 |
| Retron | III | 3 | 1 | 1 | Reverse transcriptase | Yes | 40 |
| Lamassu | – | 1 | 1 | 1 | Unknown | Yes | 16 |
| **No activity** | | | | | | | |
| AbiJ | – | 2 | 1 | 1 | Unknown | Yes | 79 |
| **Not tested** | | | | | | | |
| RloC | – | 17 | 6 | 1 | Translation stop | Yes | 42 |
| PrrC | – | 3 | 1 | 1 | Translation stop | Yes | 43 |
| Stk2[§] | – | 2 | 2 | 1 | Phosphorylation | Yes[§] | 41 |

Defense systems were omitted from the homolog counts when at least one of the defense genes was a pseudogene. Defense systems are classified according to their activity spectrum against tested streptococcal phages. [§]Pseudogene. *Abi* Abortive infection, *Ref.* Reference.

28% (74/263) of the strains. The other systems were present in less than 15% of the strains. A comparison of the prevalence of these systems in *S. thermophilus* compared to related bacterial taxa is presented in Supplementary Data 5. Most of the systems were found as one copy per genome, but although a few (AbiD and SoFic) were occasionally present twice (Table 1).

While the mode of action for some of these defense systems remains unknown, others have been characterized in other bacterial species as indicated in Table 1. Four abortive infection (Abi) systems (AbiD, AbiE, AbiH, and AbiJ), originally characterized in the LAB *Lactococcus lactis*, have also been identified in *S. thermophilus*[31]. These systems block phage replication at various stages before cell lysis, thereby preventing the spread of the infection to neighboring cells[32]. Besides the classical Abi systems, most defense systems present in *S. thermophilus* are associated with population-level protection via an Abi phenotype (Table 1). Two key strategies seem to drive Abi defense in *S. thermophilus*: depletion of NAD+ and/or ATP[16,33,34] within the infected cell (Gao19, Thoeris, and SEFIR) and non-specific DNA degradation[34-37] (Gao19, Gabija, Hachiman, and Kiwa). Signal-based defenses like CBASS[38] and Thoeris[33], which rely on intracellular signaling molecules (e.g., cyclic oligonucleotides and cyclic ADP-ribose) to activate effector proteins inducing cell death, are also present. Toxin-antitoxin systems[16,39] (AbiE and RosmerTA), retrons[40], and phosphorylation of essential cellular pathways by Stk2[41] are also found in a few strains. Lastly, RloC[42] and PrrC[43] systems, which inhibit translation following the neutralization of type I RM systems by phage proteins, were found in 6% (17/263) and 1% (3/263) of strains, respectively, suggesting that *S. thermophilus* phages may carry anti-RM proteins.

## Accessory defense systems provide resistance against dairy phages

Eighteen defense systems were experimentally tested against 16 dairy phages, including 14 representatives from the five genera that infect *S. thermophilus*, as well as two phages (*Skunavirus* p2 and *Ceduovirus* c2) that infect *Lactococcus cremoris* (Fig. 3A). The efficacy of each defense system was evaluated by measuring the efficiency of plaquing (EOP) of phages on strains that harbor the plasmid-encoded systems. The efficacy of low-copy (pTRKL2) and high-copy (pNZ123 and pTRKH2) vectors[44,45] were initially compared to identify the most suitable for assessing defense effectiveness (Fig. 3B). Overall, pTRKL2 and pTRKH2 conferred higher levels of phage resistance, particularly with Kiwa and Dodola. Notably, no phage resistance was observed for Dodola when the system was cloned into pNZ123, underscoring the critical role of plasmid selection in such experiments. This suggests that features like copy number and origin of replication significantly influence the outcomes. Based on these findings, we proceeded with the low-copy vector pTRKL2 for further cloning experiments.

The 18 distinct defense systems that were tested demonstrated significant effectiveness, achieving at least 1-log reduction in titers against one or more of the tested phages (Fig. 3C). Specifically, 17 of the 18 tested defense systems were effective against streptococcal phages, with some providing protection up to almost 7 logs (Supplementary Figs. 4 and 5). While defense systems such as Gabija, Dodola, Hachiman, and Thoeris conferred broad-spectrum protection across all tested streptococcal phages, other systems displayed more selective activity. For instance, AbiH and Borvo-like systems specifically inhibited the replication of *Vansinderenvirus* (Fig. 3C). Some defense systems, including SEFIR and RosmerTA, exhibited even narrower

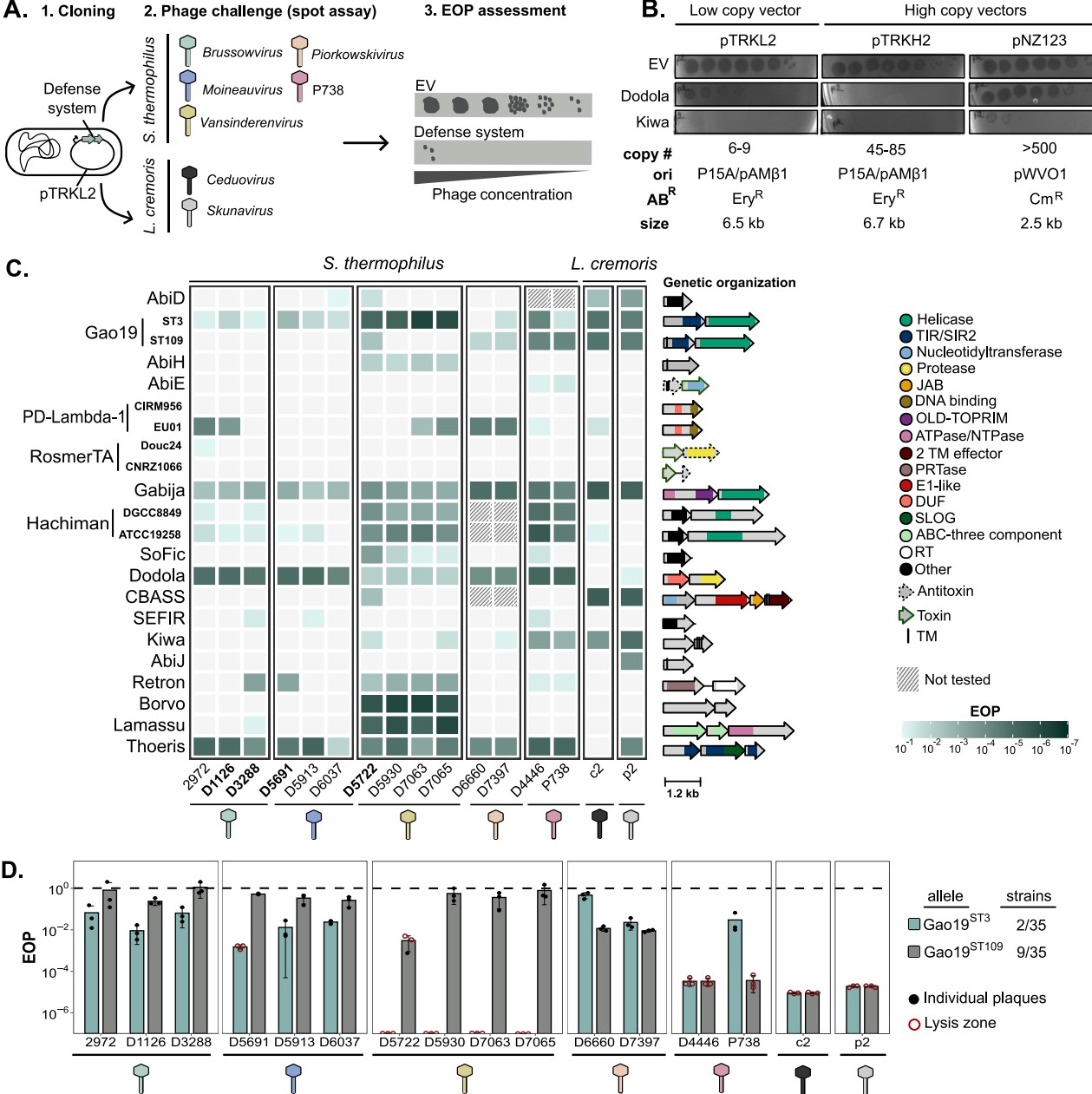

**Fig. 3 | Activity of defense systems against phages infecting *S. thermophilus* and *L. cremoris*. A** Strategy for assessing the efficiency of defense systems against streptococcal phages from five genera and lactococcal phages from two genera. Anti-phage activity was measured by evaluating the EOP of phages on the host strain containing the defense mechanism cloned on a plasmid compared to the host strain with the empty plasmid. **B** Comparison of Dodola and Kiwa defense systems efficiency against *L. cremoris* phage p2 when cloned into low-copy and high-copy vectors. **C** EOP heatmap of phages on strains carrying each defense system. Darker shades indicate stronger anti-phage protection. An EOP of $10^{-1}$ was used as the minimum threshold for anti-phage activity. Three biological replicates were done. Some defense systems were not tested against a few phages, as they are naturally present in corresponding host strains. Two homologs were tested for Gao19, PD-Lambda-1, RosmerTA, and Hachiman. Phages encoding known ACR are highlighted in bold. The top to bottom presentation of the defense systems reflects

the high to low prevalence (Fig. 1A). The genetic organization of the defense systems is shown on the right side of the heatmap. A detailed description of their functional domains is available in Supplementary Data 6. **D** Comparison of anti-phage activity of two Gao19 homologs. Gao19 was found in 35 *S. thermophilus* strains, of which 2 corresponded to Gao19ST3 and 9 to Gao19ST109. Bars show the mean EOP with error bars indicating standard deviation from biological replicates ($n = 3$). Filled circles indicate the presence of countable plaques, while hollow red circles denote lysis zones where plaques were not counted. Half circles indicate the absence of visible plaques or lysis zones even with undiluted phage lysate, showing that the defense system reduced phage levels below the limit of detection (EOP < $10^{-7}$). ABR Antibiotic resistance gene, CmR Chloramphenicol resistance, DUF Domain of unknown function, EOP Efficiency of plaquing, EryR Erythromycin resistance, EV Empty vector, ori Origin of replication, SIR Sirtuin, TM Transmembrane domain. Source data are provided as a Source Data file.

specificity, targeting only specific phages within one viral genus. This underscores the importance of testing multiple phages, including closely related ones, to uncover intra-genus specificities.

Of interest, *Brussowvirus* and *Moineauvirus*, which are the most prevalent phages in the industry, were less sensitive to the tested

defense systems, with only four and three defense systems conferring some level of protection against all the phages from these viral genera, respectively. Overall, the level of protection was also higher against other genera, with most systems providing over a 3-log reduction in phage titers for these less common genera. *L. cremoris* phages were

also effectively targeted by these defense systems originating from *S. thermophilus*, with some offering high levels of protection.

Overall, the resistance patterns observed across our tested defense systems generally exhibited higher levels of protection compared to those reported by Kelleher et al.[22]. While this difference may partly be explained by the use of different phages or defense homologs, it could also be attributed to our use of a low-copy vector plasmid, which likely enhanced the resistance conferred by systems like Gao19 and Gabija, where the same homologs were tested in both studies.

### Defense system homologs exhibit diverse activity spectra

To explore whether different defense system homologs (Table 1) offer varying protection against phages, we conducted a comparative assessment of the activity of some of these homologs (Fig. 3C, D). Gao19 (also known as the SIR2-HerA complex or Nezha) was identified in 35 strains, encompassing 12 homologs (Table 1). We compared the effectiveness of two of them, Gao19$^{ST3}$ (present in two strains) and Gao19$^{ST109}$ (present in nine strains). Despite sharing only 17.9% and 22% amino acid (aa) identity for the SIR2 and HerA proteins, respectively, both homologs had the same predicted domains (Supplementary Data 6). However, the Gao19$^{ST109}$ SIR2-like protein included a unique N-terminal transmembrane domain (Fig. 3D). While both homologs exhibited similar activity against phages infecting *L. cremoris*, differences were observed with streptococcal phages (Fig. 3D and Supplementary Fig. 6A). Gao19$^{ST3}$ provided over 6-log protection against *Vansinderenvirus* phages, whereas Gao19$^{ST109}$ achieved only a 2-log reduction, and only for phage D5722. In addition, low-level protection against *Moineauvirus* and *Brussowvirus* was observed exclusively with Gao19$^{ST3}$. Conversely, Gao19$^{ST109}$ was more effective than Gao19$^{ST3}$ against certain *Piorkowskivirus* and P738 phages.

Distantly related homologs of Hachiman (HamA and HamB subunits, sharing 15% and 17% aa identity, respectively) with similar domain annotations provided similar levels of resistance against the tested phages (Fig. 3C). In contrast, RosmerTA homologs, which were more distantly related (with toxins sharing 11% aa identity and antitoxins 6.7%), proved inefficient against most tested phages. The RosmerTA$^{CNRZ1066}$ homolog lacked the peptidase M78 domain, which is typically found in this defense system. Interestingly, two homologs of the PD-Lambda-1 system, despite sharing 98.5% aa identity, displayed divergent anti-phage activity. PD-Lambda-1$^{CIRM956}$ was completely ineffective, while PD-Lambda-1$^{EU01}$ provided a high level of resistance (>3-log reduction) against some *Brussowvirus*, *Vansinderenvirus*, and *Piorkowskivirus* phages (Fig. 3C). This emphasizes the importance of testing different homologs to evaluate the specificity of defense systems, as even minor aa changes can significantly impact their activity.

### Combining defense systems enhances immunity against phages

Given the ubiquity and the crucial role of the chromosomally-encoded CRISPR-Cas systems in generating resistant strains, we aimed to evaluate whether combining CRISPR-Cas with other defense systems could enhance protection against phages, particularly those encoding ACR proteins. A DGCC7710 strain containing a spacer in its CR1 locus that targets the tested phages (CR1-immune) was transformed with a plasmid containing one defense system (Gabija, Hachiman, or Thoeris). The effectiveness of individual and combined defenses was assessed using solid and liquid assays (Fig. 4A–C and Supplementary Fig. 7).

CR1 immunity effectively protected against phage 2972, a phage that does not encode known ACR, reducing its titer by 3-logs on plates (Fig. 4A and Supplementary Fig. 7A). Non-CRISPR defense systems provided a 3–5-log protection against this phage when used individually (Fig. 4A and Supplementary Fig. 7A). Combining CR1 immunity with either Gabija, Hachiman, or Thoeris reduced phage replication below the limit of detection, with no escape phages observed in spot

tests (Fig. 4A). Phage D5691 encodes an AcrIIA6, which enables it to completely bypass the defense provided by the CR1 system when tested on plates (Fig. 5A and Supplementary Fig. 7A). However, D5691 was sensitive to Gabija, Hachiman, and Thoeris, each of which conferred a 3–4-log protection when tested individually and hinder phage replication in CR1-immune strain (Fig. 4A and Supplementary Fig. 7A). The ability to target phages carrying other ACRs (i.e., AcrIIA3 from phage D5691 and AcrIIA5 from D1126) was also shown (Supplementary Fig. 7A). In addition, using phage D3288 (which was not targeted by the spacers in the CR1 and CR3 arrays), we showed that combining an additional defense system can neutralize phages that would otherwise evade CRISPR-Cas defenses due to the absence of a matching protospacer (Supplementary Fig. 7A).

The efficiency of CR1 combinations was further assessed in liquid medium using killing assays across a range of multiplicities of infection (MOIs), from 0.0005 to 50 (Fig. 4B, C). In the CR1-immune strain, complete protection against phage 2972 (no ACR) was only observed at MOIs below 0.05. At higher MOIs, phage 2972 still infected the CR1-immune strains, albeit with a delay, due to the emergence of CEMs. Among 20 phage plaques screened following the assay, 19 harbored a deletion in their protospacer, and one contained a point mutation, allowing evasion of CRISPR-Cas targeting (Fig. 4D).

As for the ACR-carrying phage D5691 (ACR +), it could only infect the CR1-immune strain at MOIs above 0.05, consistent with the MOI threshold required for cooperation between ACR-encoding phages to fully evade CRISPR-Cas defense[46]. Moreover, D5691 infected the CR1-immune strain more efficiently than phage 2972 (ACR-), as shown by the earlier decline in culture OD at MOIs above 0.05. Sequencing of 10 plaques revealed intact protospacer and PAM sequences, confirming that phage D5691 can bypass CR1 immunity without requiring escape mutations (Fig. 4D).

Overall, for both phages, the expression of Gabija in CR1-immune DGCC7710 enhanced resistance at MOIs above 0.005, allowing bacterial growth to reach OD levels comparable to those observed in the absence of phages in most cases (Fig. 4B). Area under the curve (AUC) analyses revealed a synergistic interaction between CRISPR-Cas and Gabija at MOIs ranging from 0.5 to 50, where their combined effect exceeded the sum of their individual protection (Fig. 4B and Supplementary Data 7). Similar synergistic effects were observed across the same MOI range when CRISPR-Cas was combined with Hachiman or Thoeris, (Fig. 4C and Supplementary Fig. 7B, C). Analysis of the escape phages following the killing assays showed that combining CRISPR-Cas with an additional defense system reduced the number of escape mutants below the detection limit (Fig. 4D). Although lysis zones were occasionally observed, attempts to isolate escape phages were unsuccessful.

Combinations of CR3 immunity with Gabija, Hachiman, or Thoeris also enhanced phage resistance, showing both additive and synergistic effects at specific MOIs (Supplementary Fig. 8). In addition, Dodola, Gabija, and Thoeris that conferred strong individual resistance (Fig. 3C) were also tested to assess their combined efficacy (Supplementary Fig. 9). In these experiments, one system (Dodola or Gabija) was chromosomally integrated into *S. thermophilus* DGCC7710, while the second was provided in trans via pTRKL2. The high individual performance of the chromosomally integrated systems, which provided near-complete protection at most tested MOIs, complicated the interpretation of the interactions. Nevertheless, additive effects were observed for Dodola with either Gabija or Thoeris (Supplementary Fig. 9A, B). An antagonistic effect was observed when Gabija was combined with Thoeris and tested at MOI 50 with phages 2972 and D5691 (Supplementary Fig. 9C).

### Multi-defense strains outperform mixed mono-defense strains

We next aimed to compare two strategies for the development of a starter culture: pyramiding, where multiple defense systems are integrated into a single strain, and mixing, where individual single-defense

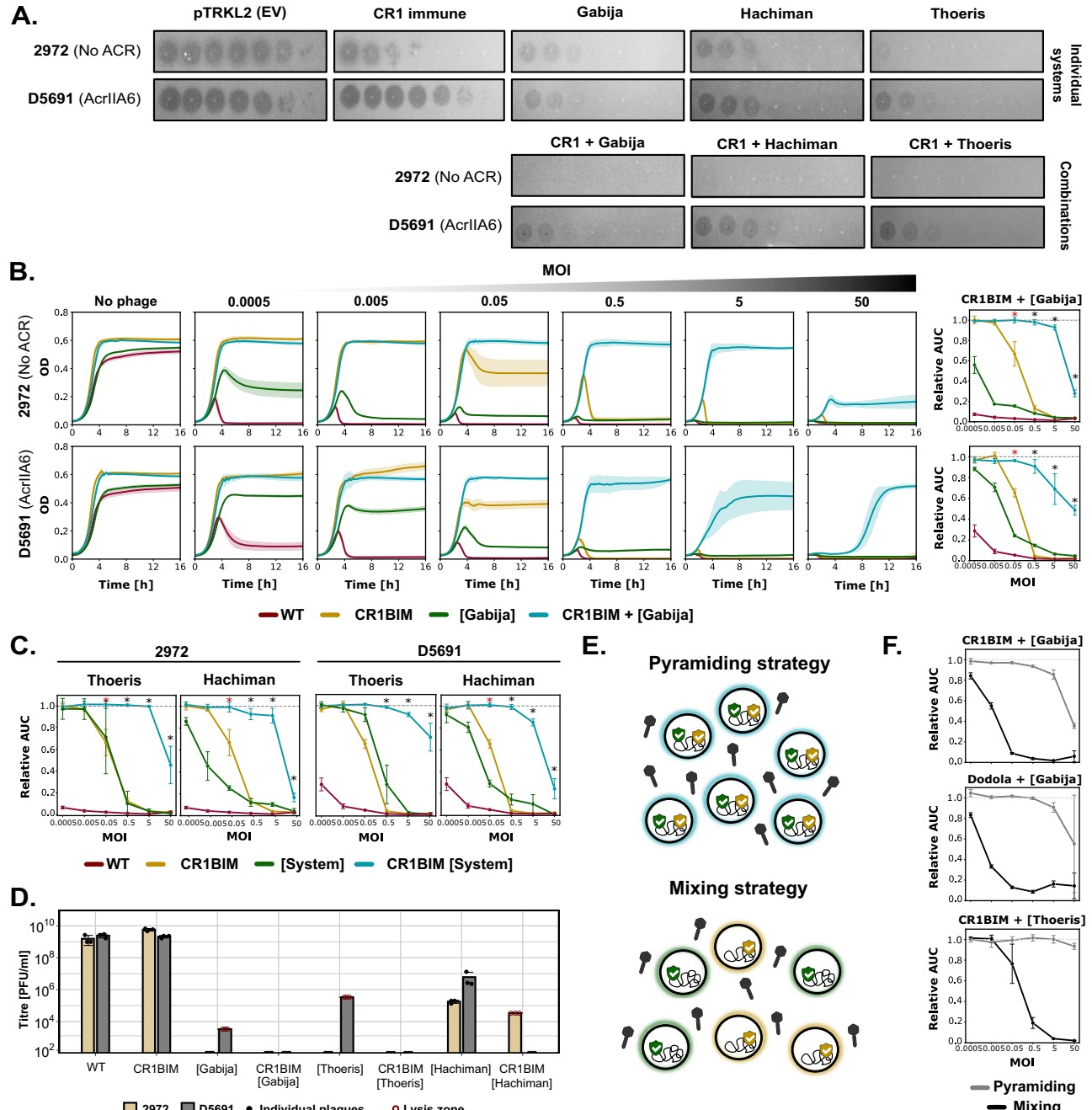

**Fig. 4 | Combining defense systems enhances phage resistance. A** CR1-immune *S. thermophilus* DGCC7710 was transformed with the pTRKL2 vector carrying either Gabija, Hachiman, or Thoeris. Strains expressing combinations of two defense systems were compared to strains carrying individual systems. Spot tests illustrate resistance profiles against phage D5691 (encoding AcrIIA6, which inhibits CR1) and phage 2972 (lacking known ACR). **B** Liquid culture assays comparing CR1 + Gabija combinations against phages 2972 and D5691 at initial MOIs ranging from 0.0005 to 50. Control strains include wild-type DGCC7710 (WT) and CR1-immune DGCC7710 (CR1BIM) carrying the empty pTRKL2 vector. The area under the curve (AUC) was calculated for each condition, and the relative AUC was determined by normalizing to the non-infected control. Synergy between defense systems is indicated by a black asterisk, and additive effects are indicated by a red asterisk. **C** Relative AUC values for combinations of CR1 with either Thoeris or Hachiman against phages 2972 and D5691. The term "System" refers to Thoeris or Hachiman as indicated. Synergy and additive effects are indicated by an asterisk color in black

and red, respectively. **D** Quantification of escape phages recovered after liquid assays at MOI 0.05 with phages 2972 and D5691 for each individual and combined defense system. Filled circles indicate the presence of countable plaques, while hollow red circles signify lysis, but plaques were not counted. **E** Schematic illustration of pyramiding (combining two defense systems in the same strain) versus mixing (co-culture of two strains, each expressing a different system). **F** Comparison of pyramiding versus mixing strategies for three combinations: CR1 + Gabija, Dodola + Gabija, and CR1 + Thoeris. For all experiments, lines or bars represent the mean value of biological replicates (*n* = 3), with shaded areas indicating confidence intervals and error bars indicating standard deviation. The statistical significance of the synergy scores was evaluated using a two-sided one-sample *t* test with Benjamini-Hochberg correction for multiple testing. A positive score significantly different from zero (*p*-value < 0.05) indicated synergy, while a score not significantly different from zero indicated an additive effect.

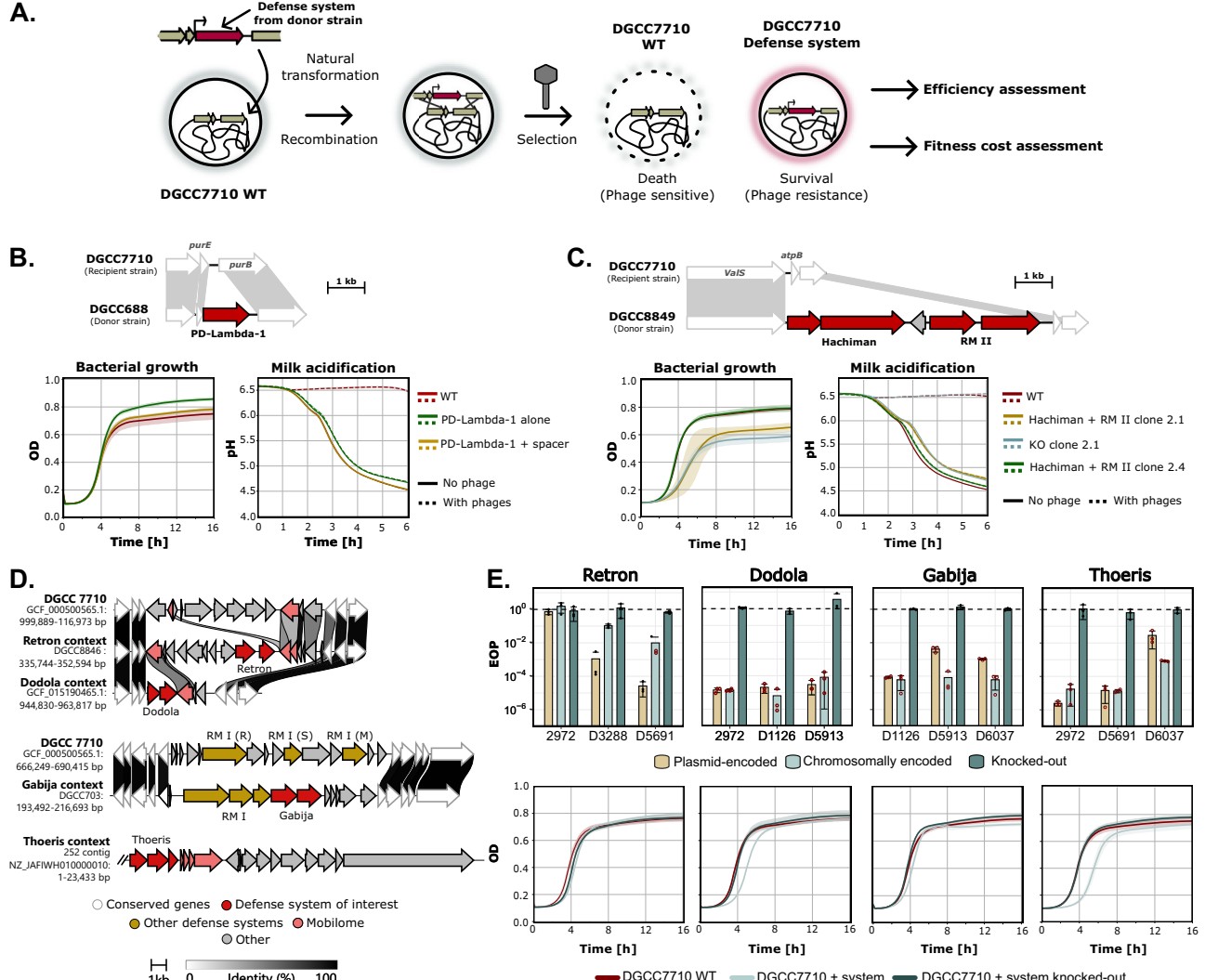

**Fig. 5 | Fitness impact of integrating an additional defense system or island into the chromosome of *S. thermophilus* DGCC7710. A** Schematic illustration of the insertion of a defense system into the chromosome of *S. thermophilus* through natural competence. A recombination template, consisting of the defense system flanked by 1 kb regions that are homologous to the upstream and downstream sequences in the recipient strain (e.g., DGCC7710), was transformed. Phages were used to select bacteria that integrated the defense system. **B, C** Assessment of the fitness cost (bacterial growth and milk acidification monitoring) following chromosomal integration of PD-Lambda-1 (**B**) or a defense island comprising Hachiman and RM II (**C**) at their native locus in DGCC7710. For bacterial growth graphs, each line represents the mean of biological replicates (*n* = 3) with shaded areas indicating confidence intervals. In milk acidification assays with phages, a phage cocktail was used at an initial MOI 0.0005. Each line shows the mean values, with shaded areas representing confidence intervals calculated from either three technical replicates

(for PD-Lambda-1, Hachiman clone 2.4, and Hachiman clone 2.1 knockout conditions) or two biological replicates, each with three technical replicates (for Hachiman clone 2.1 and DGCC7710 WT). **D** Genomic context of selected defense systems integrated at non-native loci in DGCC7710. Thoeris originates from a short contig lacking conserved neighboring genes, preventing its precise localization within the DGCC7710 genome. **E** Evaluation of defense efficiency (EOP) and fitness cost (bacterial growth monitoring) for systems integrated at non-native loci. All experiments were performed in biological replicates (*n* = 3). Bars and curves indicate mean values, with standard deviations shown as error bars and confidence intervals as shaded areas. In the EOP plots, filled circles represent conditions with countable plaques, while hollow red circles indicate lysis zones without distinct plaques. EOP Efficiency of Plaquing, WT Wild Type, OD Optical Density, RM Restriction-modification system, R-S-M Restriction-Specificity-Methylase subunits of RM type I system. Source data are provided as a Source Data file.

strains are combined in a population (Fig. 4E)[47]. To evaluate their effectiveness, we performed a killing assay with phage 2972. We tested strains carrying a chromosomal defense system (either CR1 or Dodola) alongside a plasmid-encoded system (either Gabija or Thoeris) and compared their resistance to that of a 1:1 combination of strains, each harboring only one of these systems.

Across all three combinations, differences between the two strategies were minimal at low MOI (0.0005) (Fig. 4F and Supplementary Fig. 10). However, at higher MOIs, the pyramiding strategy consistently outperformed the mixing strategy, supporting previous findings[47] that pyramiding defense systems reduces the emergence of escape mutants and enhances resistance.

## Chromosomal integration of defense systems shows no fitness cost

To use a defense system in industrial strains, it must be transferable across strains via processes such as conjugation, transduction, or natural transformation. *S. thermophilus* can perform natural transformation, which can be leveraged to chromosomally integrate a defense system[48] (Fig. 5A). Transformants that have successfully integrated the defense system can be screened by PCR following phage selection. Since maintaining multiple defense systems may impose a fitness cost[49,50], we assessed the impact of introducing additional defense systems under both laboratory and industrial conditions.

The defense system PD-Lambda-1$^{EU01}$ was selected for chromosomal integration at its native locus in the industrial strain DGCC7710 (Fig. 5B). Integrated PD-Lambda-1$^{EU01}$ provided protection against *Brussowvirus* comparable to that of the vector-encoded version (Supplementary Fig. 11A). Its role in phage resistance was further confirmed by the complete loss of protection upon removal of the integrated system. To assess fitness impact, two recombinant derivatives were analyzed: one carrying only the integrated PD-Lambda-1 system, and another that had also acquired an additional spacer during phage selection, resulting in enhanced phage resistance (Supplementary Fig. 11A). Neither clone exhibited impaired bacterial growth in LM17 medium (Fig. 5B). Milk acidification, a key industrial trait, was also evaluated for both clones and compared to the WT strain. The strain carrying only PD-Lambda-1 showed a slightly slower acidification rate ($-1.26 \times 10^{-2} \pm 5.73 \times 10^{-5}$ pH unit/min) compared to the WT ($-1.44 \times 10^{-2} \pm 1.78 \times 10^{-4}$ pH unit/min), while the acidification curve of the clone harboring both PD-Lambda-1 and the additional spacer ($-1.42 \times 10^{-2} \pm 1.45 \times 10^{-4}$ pH unit/min) was similar to WT. These results indicate that the integration of PD-Lambda-1 does not impair the strain's acidification ability. Under milk fermentation conditions, both clones maintained strong resistance to phage infection. Indeed, the addition of a phage cocktail did not impair milk acidification in either clone, in contrast to the WT strain, whose sensitivity to phages prevented acidification. These findings demonstrate that PD-Lambda-1 provides effective protection against phages in the milk environment at a native locus.

Similarly, a 7.2 kb island composed of two defense systems, Hachiman and a type II RM, was integrated into its native locus in DGCC7710 (Fig. 5C). This integrated island conferred strong phage resistance, with only lysis zones observed at the highest phage titers, far exceeding the protection provided by Hachiman alone when expressed from the pTRKL2 vector (*ca.* 1-log protection; Supplementary Fig. 11B). Growth analysis of two recombinant clones carrying the integrated island showed no significant difference for clone 2.4 compared to the WT strain, whereas clone 2.1 exhibited slower growth. However, a knockout of the defense island in clone 2.1 did not restore normal growth, indicating that the reduced growth was not due to the defense island but likely to secondary mutation(s) acquired during integration. Similarly, milk acidification assays showed a slight reduction in the acidification rate for clone 2.4 ($-1.27 \times 10^{-2} \pm 1.49 \times 10^{-4}$ pH unit/min) compared to the WT ($-1.42 \times 10^{-2} \pm 2.93 \times 10^{-4}$ pH unit/min) and a more important reduction for clone 2.1 ($-1.08 \times 10^{-2} \pm 8.84 \times 10^{-4}$ pH unit/min). This slowdown also persisted in the knockout version of clone 2.1 ($-1.15 \times 10^{-2} \pm 2.31 \times 10^{-4}$ pH unit/min), further indicating that the phenotype was unrelated to the defense island integration. Importantly, the defense island was effective in preventing phage infection in milk, a protection that was lost in the knockout clone 2.1, further confirming the efficiency of these defense systems in *S. thermophilus* environment.

Finally, we selected four additional defense systems for chromosomal integration in DGCC7710. In this strain, the native genomic loci where these systems are typically found in other strains are occupied by unrelated genes (Fig. 5D). Since our goal was to add additional defense systems (i.e., pyramiding strategy) without disrupting potentially functional or uncharacterized genes, we chose to integrate these systems into the genomic location of PD-Lambda-1. This strategy ensured that no existing genes were removed. All integrated defense systems conferred protection against phages, with most displaying comparable levels of resistance to their plasmid-encoded counterparts (Fig. 5E and Supplementary Fig. 11A). However, variations in efficacy were observed depending on specific phage-defense system combinations. For example, Retron was more effective when expressed from the pTRKL2 plasmid than when chromosomally integrated, whereas Gabija provided a 1–2-log improvement in resistance against two phages when integrated into the chromosome. Liquid assay

comparisons further revealed differences in defense efficiency between plasmid and chromosomal expressions, underscoring the influence of genomic context for the assessment of defense system activity (Supplementary Fig. 12).

Regarding fitness impact, chromosomal integration of Retron and Gabija did not affect bacterial growth. In contrast, Dodola and Thoeris caused a growth delay that was eliminated upon knockout, suggesting a metabolic burden associated with their expression at that locus. Surprisingly, all four defense systems impacted fermentation properties as evidenced by the absence of milk coagulation (Supplementary Fig. 13A–D), which is a direct result of lactic acid production and associated pH drop. Additional evaluation of milk acidification for the Dodola-integrated strain showed complete inhibition of milk acidification (Supplementary Fig. 13E). These observations suggest that integration of certain defense systems into non-native loci can impair strain fitness.

## Discussion

Preventing phage infection remains an industrial challenge. A primary strategy to control these infections involves optimizing bacterial cultures by exposing strains to phages and naturally selecting BIMs with enhanced resistance. In *S. thermophilus*, CRISPR-Cas systems are the primary drivers of BIM selection through the acquisition of novel spacers, as very few non-CRISPR BIMs have been described for this bacterium[51,52]. While CRISPR-Cas remains the most effective tool for developing phage-resistant *S. thermophilus* strains, phages can overcome this resistance by mutating their protospacers or PAM[10]. Approximately 40% of streptococcal phages were also found to encode at least one ACR protein, enabling them to bypass type II-A systems. This limitation necessitates a search for alternative defense mechanisms to develop more effective strategies for combating streptococcal phages in the dairy industry.

In many other bacteria, the primary defense mechanism against phages is the alteration of the receptor required for phage adsorption. While *S. thermophilus* BIMs with mutations in genes involved in the biosynthesis of the polysaccharide receptor have been isolated, they exhibit an altered growth phenotype that hinders their industrial use[53]. Bacteria have a multitude of intracellular defense systems capable of interrupting the viral cycle at every stage, from genome injection to host lysis[4]. Recent advances in predictive tools for identifying known defense systems in microbial genomes uncovered that, on average, bacteria encode approximately five such systems[23]. This suggested that *S. thermophilus* likely harbors additional defense mechanisms beyond the CRISPR-Cas systems. Our analysis revealed that strains encode an average of 7.5 defense systems, with CRISPR-Cas and RM systems being the most prevalent. This higher prevalence of defense systems likely underlines their necessity for *S. thermophilus* to thrive in a changing virus-containing ecosystem.

Here, we identified 21 accessory defense systems, each present in less than 30% of the strains, significantly expanding the defense repertoire of *S. thermophilus*. Interestingly, most of these accessory systems have been shown to cause the collapse of the bacterial population at high MOI, a phenotype associated with Abi systems that prevents the viral outbreak from spreading to neighboring cells[54]. Although an actual Abi mechanism has not been confirmed for several of the accessory defense systems highlighted, we speculate that *S. thermophilus* may have evolved various Abi systems as a secondary line of defense, which is activated when phages have managed to circumvent CRISPR-Cas and/or RM systems. Abi systems such as PrrC and RloC, detected in several strains, are known to be triggered when phage proteins inactivate type I RM systems[42,43].

Seventeen defense systems demonstrated variable levels of resistance against a diverse panel of streptococcal phages. Among these, *Moineauvirus* and *Brussowvirus*, the most frequently encountered genera in the industry, were the least susceptible to the tested

systems. In contrast, other less commonly isolated viral genera displayed greater sensitivity. These rarer genera, which share homologies with lactococcal and non-dairy streptococcal phages, are thought to have emerged more recently than *Moineauvirus* and *Brussowvirus*, which have been isolated for decades[55–57]. This suggests that the latter may have evolved multiple strategies to evade *S. thermophilus* defense systems, contributing to their success in the dairy environment. Notably, some defense systems are activated when they sense the presence of specific phage proteins. When phages have mutations in these activators, or if they are absent, they can bypass the defense mechanisms[58]. Structural proteins, in particular, have been implicated in the activation of several defense systems[59]. Comparative genomics of the structural modules in *S. thermophilus* phages revealed significant divergence, potentially explaining the differences in sensitivity among certain phages[60]. In addition, phages can escape defense systems by encoding counter-defense proteins[61]. We showed that ACR proteins are more prevalent in *Moineauvirus* and *Brussowvirus*, and other, not yet identified, anti-defense proteins may also be present. However, the absence of significant candidates predicted by bioinformatics tools suggests that these anti-defense proteins may differ from those known currently, many of which have been identified in phages infecting Gram-negative bacteria[62].

Given the ubiquity of CRISPR-Cas systems in *S. thermophilus* and their key role in developing phage-resistance strains, newly identified defense mechanisms will always be used in combination with CRISPR-Cas rather than as replacements. Previous research has demonstrated that CRISPR-Cas systems are compatible with type II RM systems, offering additive protection against streptococcal phages[63]. Our study further demonstrates that other DNA-degrading systems, such as Gabija and Hachiman, as well as NAD+ depletion-based systems like Thoeris, are also compatible with CRISPR-Cas, leading to increased resistance against phages, including those carrying ACRs. It is anticipated that additional defense systems will exhibit similar compatibility with CRISPR-Cas, as defense mechanisms generally work well together and can even exhibit synergistic effects[64]. Such synergy was clearly observed at several MOIs when CRISPR-Cas immunity was combined with either Gabija, Thoeris, or Hachiman. Moreover, the number of phages able to escape both defenses was often reduced below the detection limit. This observation further supports the notion that combining defense systems efficiently prevents the emergence of escape mutants.

In the case of phage 2972, which does not encode an ACR, mutations or deletions in the protospacer to bypass CRISPR interference are unlikely to affect sensitivity to an additional defense system, unless the targeted gene encodes a protein involved in activating that system. For phages like D5691, which encodes ACRs, it was previously shown with the MADS defense system that although escape mutants can arise when the system is used alone, their frequency within the population is too low to allow the cooperation required to overcome CRISPR immunity[65]. As a result, the combined activity of CRISPR-Cas with another defense system (MADS, Thoeris, Gabija, or Hachiman) makes it significantly more difficult for the ACR phage to replicate. Finally, we confirmed that the pyramiding strategy, where multiple defense systems are stacked within the same strain, provides stronger phage resistance compared to the mixing strategy, where strains with individual defense systems are combined[47]. This was observed not only for combinations involving CRISPR-Cas, but also with non-CRISPR defense systems.

While stacking multiple defense systems can broaden the variety of targeted phages or help counter phages that have evolved escape mechanisms, the accumulation of numerous defenses may have a fitness cost in the absence of phage pressure, due to factors such as metabolic burden or risks of self-targeting[49,66]. Here, we successfully mobilized either a single defense system (PD-Lambda-1) or an entire defense island (Hachiman + RM II) into their native chromosomal loci in an industrial strain, resulting in robust phage resistance. The presence of these defense systems did not affect bacterial growth or milk acidification rates. In contrast, the integration of four other defense systems into non-native chromosomal loci led to a different outcome. While most of these insertions had no or only moderate effects on bacterial growth, all abolished the strain's ability to ferment milk. Although we were unable to test these systems in their native loci, this observation suggests that the genomic context can significantly influence the fitness costs associated with defense system integration.

More broadly, this study underscores the importance of investigating defense systems within their native host and genomic context. While defense systems are often assessed in heterologous hosts such as *E. coli* or *Bacillus subtilis* under controlled laboratory conditions, such setups may not accurately reflect their activity or fitness impact in more natural environments. Most characterized defense systems are known to exhibit relatively narrow activity, typically targeting specific phages within a given genus[67]. In contrast, our results demonstrate that *S. thermophilus* defenses can be highly specific, yet also display broader activity, effectively targeting phages across all genera known to infect this species. In addition, we showed that the phage resistance provided by some defense systems is functional in milk, the natural environment of *S. thermophilus*. Defense systems are often tested in trans, expressed from plasmids with varying copy numbers, even when these systems are typically encoded in the chromosome. While low-copy number plasmids are commonly used, we observed differences in defense efficiency depending on whether the system is expressed from a low-copy plasmid or integrated into the chromosome. Importantly, this was also evident in liquid culture experiments conducted at various MOI, which are increasingly used to assess potential Abi phenotypes[54]. These observations highlight the importance of studying defense mechanisms under their native promoters and genomic contexts to understand their activity and functionality in the most relevant way.

In conclusion, developing phage-resistant *S. thermophilus* strains is crucial for the success and sustainability of the milk fermentation industry. Our research highlights the importance of understanding and leveraging the defense mechanisms inherent to this bacterium. By strategically combining CRISPR-Cas systems with other defense systems, we can create a robust and multi-layered defense to combat phages. This comprehensive strategy promises to enhance the resilience of starter strains, ensuring the high quality of fermented dairy products.

## Methods
### Defense system prediction
*S. thermophilus* genomes ($N = 263$) were downloaded from the NCBI Reference Sequence (RefSeq) database in March 2023 (Supplementary Data 1). Defense systems were predicted using PADLOC[24] v2.0.0 and DefenseFinder[23] v1.2.2. Predictions labeled as "other", "cas_a-daptation", "cas_cluster", or candidate systems were excluded as they did not represent complete or experimentally validated systems. Results from both prediction tools were combined using an in-house script. After manual inspection, the defense systems Viperin_solo and PD-T4-6 were excluded from the analysis as they appeared to be wrong predictions. The same genes that matched the closely related PD-T7-2 and Gao19 systems[68] were combined into a single system, denoted as Gao19. Similarly, Abi2 and AbiD, which mostly match the same genes, were clustered as AbiD systems. Type II and type IIG RM systems were grouped into RM type II. One CRISPR II-A was incorrectly predicted as CRISPR II-C and was manually corrected. The detailed list of identified defense systems can be found in Supplementary Data 2 and 3. A core proteome species-phylogeny was

generated with OrthoFinder[69], and functional analysis of defense systems was done with InterPro[70].

## Plasmid and prophage predictions

Plasmids were identified in incomplete genomes using the PlasmidFinder database with Abricate v1.0.0 (https://github.com/tseemann/abricate). Prophages were identified with Phigaro v2.4.0, and hits greater than 10 kb were confirmed using PHASTEST webserver[71].

## Restriction site analysis

RM proteins were submitted to the REBASE database (http://rebase.neb.com/rebase/) to predict restriction sites (identity threshold of 75%). For type I, only the specificity protein was used, whereas both the methylase and endonucleases were used for types II and III. Detailed information for each RM system is provided in Supplementary Data 8. To evaluate the avoidance of restriction sites, the genomes of 191 phages (116 *Moineauvirus*, 46 *Brussowvirus*, 14 *Piorkowskivirus*, 13 *Vansinderenvirus*, and 2 P738-like phages) that infect *S. thermophilus* were downloaded from NCBI in June 2023 (Supplementary Data 4). Predicted sites for types II and III RMs were counted in each genome (i.e., observed number of restriction sites). The expected number of restriction sites was estimated using a Markov-immediate neighbor dependence model[72]. Briefly, the frequency of each nucleotide and dinucleotide was calculated for each genome. For example, the frequency for GATC ($f_{GATC}$), was calculated as:

$$f_{GATC} = f(G|N) \times f(A|G) \times f(T|A) \times f(C|T) = \frac{f_G \times f_{GA} \times f_{AT} \times f_{TC}}{f_G \times f_A \times f_T}$$
$$= \frac{f_{GA} \times f_{AT} \times f_{TC}}{f_A \times f_T} \qquad (1)$$

where N represents any nucleotide, and, for example, $f(A|G)$ is the frequency of observing an A preceded by a G.

The expected number of restriction sites in each genome was obtained by multiplying the corresponding frequency by the genome length. Restriction site avoidance was calculated using the ratio between the observed and expected numbers of sites. Underrepresented sites had ratios < 0.75, and overrepresented sites had ratios > 1.25.

## ACR and methyltransferase prediction in phage genomes

ACRs proteins were predicted by dbAPIS[62] with a threshold for *e*-value relevance set at 1e⁻³⁰. Methyltransferases were identified by retrieving phage proteins with annotations containing "methyltransferase" or "methylase" and confirming their methyltransferase function using the REBASE database and InterPro[73].

## Spacer-protospacer analysis

Spacer sequences from the CR1 locus (i.e., the most ubiquitous and active in spacer acquisition) were extracted from the 263 *S. thermophilus* genomes analyzed in this study. These spacers were compared against a custom database of the 191 phage genomes using BLASTn to identify spacer matches with 100% sequence identity to protospacers in each phage genome. Of note, P738 phages (*n* = 2) were not included as none of the analyzed strains contained matching spacers targeting these phages.

## Bacterial growth conditions

Bacteria and phages used in this study are listed in Supplementary Data 9. All strains were grown overnight unless stated otherwise. *Escherichia coli* was grown at 37 °C with shaking in Brain Heart Infusion (BHI, Difco) medium. *S. thermophilus* was grown in 0.5 x M17 (Nutri-Bact) supplemented with 0.5% w/v lactose (LM17) and incubated at 37 °C without shaking. *L. cremoris* was grown in M17 with 0.5% w/v glucose (GM17) and incubated at 30 °C without shaking. For solid media, 1.5% agar (Laboratoire Mat) was added, and plates were incubated at 37 °C for *E. coli*, 42 °C for *S. thermophilus*, and 30 °C for *L. cremoris*. Erythromycin was added to the media, when necessary, to maintain pTRKL2 or pTRKH2 vectors, at final concentrations of 150 µg/ml for *E. coli* and 2.5 µg/ml or 5 µg/ml for *S. thermophilus* and *L. cremoris*, respectively Chloramphenicol was used at final concentrations of 20 µg/ml and 5 µg/ml to maintain the pNZ123 vector in *E. coli* and *L. cremoris*, respectively.

## Construction of defense system-encoding plasmids

Some of the examined defense systems were PCR amplified from our bacterial collection using primers listed in Supplementary Data 10 and the Q5 High-Fidelity DNA Polymerase (NEB). Other defense systems were synthesized at Bio Basic or Integrated DNA Technologies. Each system was then cloned into the low-copy vector pTRKL2[44] (and high-copy vectors pNZ123[45] and pTRKH2[44] for Kiwa and Dodola) via Gibson assembly. Chemically competent *E. coli* NEB5α were then transformed with the constructed plasmids. PCR screenings were performed with Taq DNA polymerase (Bio Basic), positive clones were verified by Sanger sequencing, and plasmids were extracted using the QIAprep Spin Miniprep Kit (Qiagen). Plasmids were then electroporated into *L. cremoris* MG1363[74]. For *S. thermophilus*, most plasmids were electroporated into relevant strains as described previously[75] with the following modifications: after the 2 h incubation at 42 °C, the transformation mix was transferred to 10 ml LM17 supplemented with erythromycin, incubated overnight at 42 °C, and plated the following day. For plasmids/strains that do not readily electroporate, plasmids were introduced by natural competence using a protocol described elsewhere[48], except that competent cells were grown in milk. Prior to transformation, we made sure that the tested defense system was not naturally present in the bacterial strains using PADLOC and DefenseFinder. The correct sequences for the defense systems found in the transformants were verified by Sanger sequencing.

To evaluate the combinations of other systems with CRISPR-Cas, plasmids encoding Gabija, Hachiman, or Thoeris were transformed into either CR1-immune DGCC7710, whose CR1 locus contained a spacer (5'-AAGTAGCCATACAAGAAGATGGATCAGCA-3') targeting *orf20* of phage 2972 and its homologs, or into CR3-immune DGCC7710, whose CR3 locus contained a spacer (5'-CTGATG-GAACCTGGCCACTGCAACCACGAC-3') targeting the same *orf20* and its homologs.

## Assessment of defense system efficiency

The effectiveness of each defense system was evaluated through spot tests using phage lysates with titers of *ca*. 10⁸–10⁹ PFU/ml. Briefly, molten agar was mixed with 15% of overnight culture and poured onto 1% agar M17 plates containing 10 mM CaCl₂. Next, 3 µl drops of serially diluted (tenfold) lysate were spotted onto the plates and let dry before incubation. For *S. thermophilus*, the molten agar contained LM17, 0.5% glycine, 0.4% agarose, and 0.1% milk. For *L. cremoris*, the molten agar was made of GM17, 0.75% agar, 10 mM CaCl₂. Phage titers were evaluated after overnight incubation. The EOP was calculated as the ratio of the phage titer in the presence of the defense system to that obtained with the strain carrying the empty vector. Statistical significance of the defense system resistance was assessed in R using a one-sample, two-sided *t* test to evaluate whether the mean EOP was significantly less than 1, with an EOP of 1 indicating no phage protection. Multiple testing correction was applied using the Benjamini-Hochberg method. A defense system was considered effective when showing both statistical significance and at least a 1-log reduction in EOP.

## Chromosomal insertion of defense systems

PD-Lambda-1 (donor strain DGCC688) or a defense island composed of Hachiman and an RM type II system (donor strain DGCC8849) were integrated into their native location in the chromosome of *S. thermophilus* DGCC7710, between the genes CW339_RS00280/CW339_RS00285 and CW339_RS02370/CW339_RS02375, respectively. The defense system or island, together with 1 kb upstream (thus including the native promotor) and downstream flanking regions, were amplified by PCR using the Q5 DNA polymerase for the 3.5 kb region comprising PD-Lambda-1 or the Phusion DNA polymerase (NEB) for the 9.3 kb amplicon comprising the defense island. Then, DGCC7710 was transformed with 10 µg of PCR template via natural competence. The flanking regions of the donor and recipient (DGCC7710) strains, showing high nucleotide sequence identity, enabled recombination in the native chromosomal location. The transformation mix was added to LM17 and incubated at 42 °C. The following day, 500 µl of an overnight culture were mixed with 3 ml molten agar and 100 µl of phages (*ca.* $10^8$ PFU/ml) to select for cells that had integrated the defense system or island using a 1:1 mixture of phages D1126 and 2972 or only D5691, respectively. Other defense systems (Gabija, Thoeris, Retron, and Dodola) could not be integrated into their native location and were instead inserted in the location of PD-Lambda-1 (between genes CW339_RS00280 and CW339_RS00285). A recombination template composed of the defense system and its 1-kb upstream (containing the native promotor) and downstream flanking regions of the insertion site in DGCC7710 was synthesized at GenScript for each defense system. Transformation and selection (with phage D5691) were performed as described above.

The genome insertion and the absence of new spacers in the CR1 and CR3 loci were verified by PCR and Sanger sequencing. To confirm that the observed anti-phage activity was caused by the integrated defense system/island and not secondary mutations that would have been selected during the phage selection step, the defense systems/island were also knocked out from the transformants using a PCR template containing the insertion region of the WT recipient strain. Knockout mutants were confirmed via PCR screening.

## Killing curve assay

Overnight cultures were adjusted to an OD of 0.2 (*ca.* $10^7$ CFU/ml) in LM17 supplemented with 10 mM CaCl₂, and 40 µl were transferred to each well of a 96-well plate, which contained 140 µl of LM17 supplemented with 10 mM CaCl₂. Phage lysates (2972 or D5691) were serially diluted ten times, and 20 µl were added to test MOI ranging from 0.005 to 50. Plates were incubated for 16 h at 37 °C, with $OD_{600}$ measurements taken every 10 min after orbital shaking for 10 s. Each curve represents the mean of three biological replicates. After 20 h, the plate was centrifuged ($2200 \times g$, 5 min), and 100 µl of the supernatant for MOI 0.05 was recovered, diluted, and spotted to assess the titer of escape mutants. The Area Under the Curve (AUC) for each condition (i.e., phage, MOI, defense system) was calculated using the function trapz, designed to compute the definite integral of a set of values using the trapezoidal rule. AUCs were normalized to the AUC of the uninfected control of the same strain (relative AUC). Synergy between combined defense systems was evaluated by calculating the difference between the AUC of the combination and the sum of the AUC values for the CRISPR defense alone and the additional defense system alone. Synergy scores were calculated from three independent replicates. The statistical significance of the synergy scores were evaluated using a two-sided one-sample *t* test, testing whether the mean differed significantly from zero, and multiple testing corrections (Benjamini-Hochberg) were applied. A positive score indicates a synergistic effect, a null score reflects an additive effect of the two defense systems, and a negative score indicates that the combined

systems do not provide enhanced resistance compared to either system used alone or act antagonistically (Supplementary Data 7).

## Fitness cost evaluation

The impact on bacterial growth of the chromosomal integration of a defense system was determined through liquid assays. O/N cultures of the bacterial strains were diluted to OD 0.2 in LM17, and 20 µl were transferred to 96 well plate, each well containing 180 µl of LM17, to monitor the $OD_{600}$ at 37 °C for 16 h. Milk coagulation assays were performed by inoculating 1% of bacterial O/N culture into 10% reconstituted non-fat dry milk (NFDM) supplemented with 0.001% bromocresol purple. Samples were incubated overnight at 42 °C. Bromocresol purple is a pH indicator that shifts from purple/blue to yellow as the pH decreases, enabling visual assessment of acid production.

Milk acidification assay was performed as described in Seiler et al.[76] with minor modifications. Briefly, O/N culture of each strain in 11% NFDM were transferred into M17 supplemented with 0.5% sucrose. After the cultures reached an $OD_{600}$ of ca. 0.5, they were inoculated at 0.75% into activity milk (2% fat) containing 1% sodium formate. Samples were kept in a 40 °C water bath O/N with pH probes and CINAC software (KPM Analytic) monitoring the acidification every three minutes. To evaluate defense system activity in the presence of phages, a phage cocktail was added at an initial MOI of 0.0005: a 1:1 mixture of phages D1126 and 2972 for DGCC7710::PD-Lambda-1, and a 1:1:1 mixture of D1126, 2972, and D5691 for DGCC7710::Hachiman + RM. Conditions related to PD-Lambda-1 derivatives, Hachiman clone 2.1 knockout, and Hachiman clone 2.4 were run in technical triplicates, while DGCC7710 WT control and Hachiman clone 2.1 were done in two biological replicates each, including three technical replicates.

## Reporting summary

Further information on research design is available in the Nature Portfolio Reporting Summary linked to this article.

## Data availability

Bacterial and phage genomes were downloaded from RefSeq (https://www.ncbi.nlm.nih.gov/refseq/), and their accession numbers are available in Supplementary Data 1 and 4. All other necessary data supporting this study are included in the Supplementary Files. Source data are provided in this paper.

## Code availability

This article does not report original code.

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

## Acknowledgements

A.L. was supported by a postdoctoral fellowship from Wallonie-Bruxelles International under the grant SUB/2022/554282. S.M. acknowledges funding from the Natural Sciences and Engineering Research Council of Canada (Discovery Program). S.M. holds the Canada Research Chair in Bacteriophages. The authors extend their gratitude to Vincent Somerville for assistance with phylogenetic tree generation, to Carlee Morency for help with the AbiD experiments and statistics. In addition, we acknowledge the support of two FRQNT networks, PROTEO (https://doi.org/10.69777/341121) and Op + Lait, for providing the opportunity to present this work at international conferences. We also thank Crayon-Bleu for editorial assistance.

## Author contributions

A.L., D.A.R. and S.M. conceived the study. A.L., G.M.R. and S.M. designed the experiments. A.L., J.L. and A.M.M. conducted the experiments. D.M. and P.H. provided feedback on experiments. A.L. and P.H. performed data analysis. A.L. wrote the first draft of the manuscript. All authors edited and approved of the manuscript.

## Competing interests

D.A.R., P.H., G.M.R. and S.M. are co-inventors on patent(s) or patent application(s) related to ACR and their various uses (US11732251, US11530405). The remaining authors declare no competing interests.
