## [Transparent Peer Review file · Nature Communications]

Strengthening Phage Resistance of *Streptococcus thermophilus* by Leveraging Complementary Defense Systems

Corresponding Author: Professor Sylvain Moineau

Version 0:

Reviewer comments:

Reviewer #1

(Remarks to the Author)

In this manuscript Leprince et al. significantly extend our understanding of antiphage defenses in *S. thermophilus*, an important lactic acid bacterium of the dairy and fermentation industries. They show that although St is dominated by CR and RM systems that are well known, a deep bioinformatic dive into the genomes of this genus reveal other recently identified and obscure antiphage defense systems. Interestingly, they show that plasmid copy number and possibly plasmid replication can influence the efficacy of certain antiphage defense systems. This is an important finding as many in the field rely on ectopic expression of defense systems, often in heterologous hosts, to test phage defense. This argues that not only should native hosts be used, but the choice of plasmid should be considered. Finally, they show that they can achieve synergy with CR systems and other antiphage defenses that are more potent for phage protection than the individual systems alone. This is a well written manuscript that was easy to follow.

1. The term "Resistome" is used throughout. This term is typically associated with antibiotic resistance. Perhaps using phage-resistome would help avoid any confusion.
2. Can the authors comment on the rare antiphage defense systems found in the St genomes (such as Theoris, Lamassu, etc.). Are such systems also uncommon in other streptococci, the Firmicutes as a whole, or is this a feature of St? Why would these systems be less prevalent compared to others?
3. Can the authors comment on how they determined that no known anti-RM proteins were found in their phages? What anti-RM proteins were queried and does this include more recent anti-RM proteins that have been identified?
4. In fig 4A when screening for recombinants that are phage resistant, how do the authors confirm that resistance is not driven by selection for cell wall modification/receptor mutants despite successful identification of homologous recombination of the system?
5. It appears that in the combination experiments only CR systems were paired with non-CR systems? What about synergy between RM and non-RM systems, or more uncommon systems with each other? Is synergy exclusive to CR or more widespread.
6. In terms of synergy, could the authors provide more insight into how some of these synergies might be happening mechanistically? I am not proposing additional experiments, just some comments on how they view this phenomenon happening.

Reviewer #2

(Remarks to the Author)

Summary

Leprince et al. explore the phage resistance mechanisms in *Streptococcus thermophilus* by examining the bacterium's defense systems beyond the well-characterized CRISPR-Cas and restriction-modification (RM) systems. Using bioinformatics tools (PADLOC and DefenseFinder), the authors analyzed the genomes of 263 *S. thermophilus* strains and identified 28 defense systems. Experimental validation of 17 systems against 14 phages revealed varying levels of resistance. The study also demonstrates that combining certain defense systems can provide synergistic protection, particularly against phages encoding anti-CRISPR proteins, with potential applications for developing more robust industrial *S. thermophilus* strains. Overall, the manuscript is presented very clearly, with a focused story and nice figures, but the study does not constitute a significant advancement to the field in its current state.

Review

The manuscript spends considerable time cataloguing the defense systems present in *Streptococcus thermophilus* strains using PADLOC and DefenseFinder. While a summary of the systems in these strains is presented clearly, much of this content is descriptive rather than transformative. The generation of such a dataset is trivial using these tools, and both resources already provide pre-computed databases of the defense systems in hundreds of *S. thermophilus* strains. The identification of 21 defense systems, although not explicitly explored in *S. thermophilus* previously, only includes systems that are already characterized in other bacteria and adds little theoretical or mechanistic understanding to the broader defense system field. In contrast, the initial development of DefenseFinder (published in *Nature Communications* >2 years ago) provided a significant advancement towards understanding the distribution of defense systems in all bacteria, not just a specific species.

The screening of systems in industrially relevant strains of *S. thermophilus* and *L. cremoris* is a refreshing detour from the usual workhorses of defense system research (*E. coli* and *B. subtilis*). Expanding the known defense profile of these systems beyond coliphages is valuable to the field. However, the authors do not advance our understanding of how these systems work. Comparing these streptococcal and lactococcal phages against their coliphage counterparts may provide some insight into what is triggering these systems, but the authors do not explore this.

The demonstration of activity for chromosomally encoded defenses is also important to the wider field of defense system research, as they are often only tested in the context of multicopy plasmids. If the authors were able to demonstrate the efficacy of more types of defense systems in a chromosomal context, this would enhance the significance of the study for a broader audience.

The authors discuss a "pyramiding strategy," wherein multiple defense systems are stacked in a particular strain to effectively prevent phage escape. While the idea is generally held that there is negative selection for defense systems in the absence of phage infection pressure because of a fitness cost to maintaining these systems, some bacteria naturally harbor 50 (or more) different systems at a time. The present study would benefit from an experimental validation of the fitness costs of stacking defenses in this specific context. By assessing the impact on key industrial traits, the authors could provide a more practical assessment of the viability of this approach, adding relevance of their findings for industrial applications (which appears to be the goal for this work).

The main thesis of the study, that phage resistance is strengthened by the co-expression of complementary defense systems is already a well-established phenomenon. Where our understanding is lacking is why and how certain combinations of systems provide a synergistic effect, which the current work does not explore. Understanding the biology behind these observations will be crucial for effectively developing enhanced industrial strains. The observation that CRISPR-Cas + another defense is better than another defense alone in the presence of an anti-CRISPR is particularly interesting, but there is no evidence as to why this is happening? Does the addition of another defense system prevent the production of the anti-CRISPR, allowing normal levels of CRISPR-Cas protection? Is there physical interaction or signaling between CRISPR-Cas and the additional system? Is the expression of multiple toxic components increasing the general stress response of the host, limiting infection?

Overall, the manuscript is presented very clearly, with a consistent and easy to follow throughline and high-quality figures. The inclusion of raw tabular data in the supplementary material makes the bioinformatics portion of the work easily verifiable. The identification of defenses and anti-defenses in *S. thermophilus* and their phages is interesting from an industrial perspective but lacks in the theoretical or mechanistic understanding that it brings to the wider field. Further exploration into the mechanisms of *S. thermophilus*-encoded systems, the mechanisms of synergy between systems and/or assessment of the real-world viability of engineering industrial strains through the addition of defenses, would greatly enhance the significance of this paper. In its current state, it may be better suited to a more-specialized journal.

Additional points:

L20 & throughout: "Resistome" is a vague term that is often used to describe antibiotic resistance rather than phage defense, consider re-phrasing to "phage resistome", "defensome" or something similar.

Fig1C: The scale for branch length is missing a description of the unit being represented (nucleotide substitutions per site?).

L188: In the statement "This co-occurrence suggests a complementary role, potentially targeting different phages and enhancing the global defense of a given strain", it is unclear what is meant by "This co-occurrence". Initially it appears to refer to a co-occurrence between AbiD and SoFic, but it seems these systems are mutually exclusive in *S. thermophilus*. If it is instead saying that multiple homologues of these systems being present in the same strain suggests that they are targeting different phages, then this is rather speculative when we don't know how these systems work, and the similarity of these homologues has not been compared.

L216: The copy number of pNZ123 is incredibly high, far beyond what is standard in the field for testing the efficacy of defense systems. Considering that most of the systems identified in *S. thermophilus* were chromosomally encoded

(suggesting a natural low copy number) and that many defense systems are toxic when overexpressed, why was this vector considered in the first place?

L222: It is not clear that “all tested defense systems” provided at least a 1-log reduction in plaquing, for example PD-lambda-1 CIRM956 and RosmerTA CNRZ1066 did not.

L226: Phage replication may have been reduced below the limit of detection in these specific experimental conditions, but it is not necessarily “abolished”.

L227: The apparent broad-spectrum defense provided by some defense systems can often be attributed to the toxicity of these systems when expressed in a heterologous host rather than anti-phage activity per-se. Do the authors see any growth defects in the strains transformed with these systems?

L248: It is unclear whether defense system homologues are exhibiting different defense profiles due to inherent differences in the defense systems themselves or different compatibilities with the host strain, can the authors comment on this?

L287: Why were only these two systems tested? Were more tested but not effective?

L226: See the previous point regarding the claim that defense is “abolished”.

L307 & throughout: “X was transformed into Y” is incorrect phrasing, it should be “Y was transformed with X”.

L308: “Has a one spacer”?

FigS3: Typically, plaque forming units per mL are reported for these types of phage assays (sometimes in addition to the efficiency of plaquing data shown here), and the test for significance is between the difference in plaque forming units for the control (no defense) and the strain with defense. While change in EOP is useful for visualization, it can also obfuscate some of the underlying data that may change the interpretation of the results, so PFU should also be shown. The type of t-test performed here also requires an adjustment for multiple comparisons if each defense system is being compared to the same control. While a reduction in plaquing is often obvious without the statistics, it is important that statistics are presented clearly and correctly if they are to be included. Error bars should also be shown (as in FigS4A).

FigS3: The distinction between countable plaques and ‘fuzzy’ zones of lysis is a nice addition to the data, but it’s a little hard to differentiate between the filled/hollow circles, could these be represented in a clearer way, perhaps different colors?

L373: Population collapse at high MOI is often attributed to many systems said to have an “abortive infection” response. However, this collapse could also be explained by phage-related damage or overcoming defense at high MOIs, rather than activity of the system itself. This is unfortunately almost never differentiated, but speculation about the mechanism and evolution of such systems should be made carefully (see Afraim & Eldar, 2023; <https://doi.org/10.1016/j.tim.2023.05.002>).

L482: e^{-30} is confusing (incorrect?) notation, consider changing to 1×10^{-30} or $1e^{-30}$ if this is what is meant.

L567 & throughout: Explanation of the statistical tests performed are lacking. I.e. what were the tests comparing? Were corrections for multiple testing applied where appropriate?

Reviewer #3

(Remarks to the Author)

In this manuscript, the authors make a significant contribution to our understanding of anti-phage defense systems in *S. thermophilus*. They identify novel defense systems and assess their anti-phage activity *in vivo*, incorporating a broader range of *S. thermophilus* strains. This research is particularly valuable for those studying microbial defense mechanisms.

The authors also describe an intriguing phenomenon: certain defense systems show enhanced efficiency when integrated into the chromosome or expressed via low-copy plasmids. Additionally, they demonstrate the synergistic effects of combining some of these defense systems with the CRISPR system. Overall, this study not only deepens our understanding of anti-phage defenses in *S. thermophilus* but also opens up new possibilities for developing more robust industrial strains.

I have a few specific comments:

1. The finding that defense systems are more efficient when integrated into the chromosome or expressed from low-copy plasmids is particularly interesting. Could the authors provide more explanation on this? Are these differences in efficiency related to variations in protein expression levels?
2. While the synergistic effects between the CRISPR system and other anti-phage defenses were explored, the study hints at the potential for developing more resilient industrial strains of *S. thermophilus*. Given the broad-spectrum activity observed in the Gabija, Dodola, and Thoeris systems, it would be interesting to determine whether these systems work synergistically or antagonistically in combination.
3. To enhance reader comprehension, it would be helpful to include a detailed description of the genetic composition and functional domains of these defense systems. This would provide clearer insights into how they function.
4. Lastly, there is a minor typographical error on line 253: “SIR2-Her system” should be corrected to “SIR2-HerA.”

Reviewer #4

(Remarks to the Author)
Comments to the authors:

This manuscript provides a comprehensive analysis of the genomes of 263 *Streptococcus thermophilus* strains, identifying 28 distinct phage-resistant defense systems. These defense systems were detected using two computational tools: DefenseFinder and PADLOC. Furthermore, 17 systems and several combined systems were also experimentally validated. The methodology sounds good. This significantly expands our understanding of the phage-resistance mechanisms in this species. The study offers a valuable resource, giving a broad perspective on the diverse array of defense systems present. The complementary nature of these defense systems suggests the potential to develop more robust industrial strains, which would be of great interest to the dairy industry. Overall, this work represents an important contribution to the field.

Major concerns:

--The composition or genetic architectures of the defence systems reported in this study should be drawn for better illustration and understanding. See the Figure 2D in the reference "Diverse enzymatic activities mediate antiviral immunity in prokaryotes. *Science*. 2020 Aug 28;369(6507):1077-1084" and Table 1 of "Systematic discovery of antiphage defence systems in the microbial pangenome. *Science*. 2018 Mar 2;359(6379):eaar4120."

--The protospacer sequences in phage genomes and the corresponding spacer sequences in bacterial CRISPR arrays also represent the evolutionary "arms race" between phages and their bacterial hosts. In the Results section "Streptococcal phages have strategies to counter CRISPR-Cas and RM defenses", the authors should further investigate the patterns and dynamics of these protospacer-spacer interactions to provide more insights into the dynamic co-evolutionary processes shaping the phage-host relationship.

As an example, using the CRISPRimmunity tool, we analyzed the *Streptococcus thermophilus* strain ST3 and found 2 type II-A CRISPR-Cas systems with a total of 40 spacers matching protospacers on 216 *Streptococcus* phage genomes. The detailed information can be accessed via the following link:

http://www.microbiome-bigdata.com/CRISPRimmunity/index/datas_crisprimmunity/complete_strain/GCF_002286255.1_ASM228625v1

--Although the study demonstrated the benefits of combining multiple defence systems, it did not look in depth at the potential costs of such 'pyramid' stacking of defences, such as the reduction in bacterial growth and fermentation performance. In addition, the performance of the pyramiding strategy, where multiple defense systems are stacked within the same strain, and the mixing strategy or cocktail of strains of individual defense systems, need to be further compared and weighed.

Minor concerns:

--The study mainly focused on the phage resistance defence system of *Streptococcus thermophilus* (*Streptococcus lactis*) but did not address other important dairy fermenting bacteria such as *Lactococcus lactis*. This limits the general applicability of the results.

--Although the study systematically analysed defense systems on the genomes of 263 *S. thermophilus* strains using DefenseFinder and PADLOC and identified 28 known types of defence systems, it is likely that there are still some novel defence systems that have not been identified. Further exploration may still be needed.

Overall, this is an excellent piece of work that has important implications for a deeper understanding of the coevolutionary relationship between bacteria and phages and for the development of more robust industrial strains. But there is still a need to further expand the scope of the study, validate the performance of the combined defence systems for industrial applications, and evaluate the potential costs of multiple defence systems in order to achieve a more comprehensive and reliable solution.

Version 1:

Reviewer comments:

Reviewer #1

(Remarks to the Author)
The authors have addressed my prior comments adequately. I have not further comments.

Reviewer #2

(Remarks to the Author)
The authors have made a significant effort to address reviewer comments. The additional clarifications and experiments in this version of the manuscript have further strengthened the work. The presentation of the data and methodology is overall

very clear. No further comments.

Reviewer #3

(Remarks to the Author)

The authors have thoroughly addressed all of my concerns.

Reviewer #4

(Remarks to the Author)

The authors have fully addressed major/minor comments through adding new figures/tables (Fig 3C, Table S4-5, Fig 4F, 5B-D); additional bioinformatic analysis (spacer-protospacer matching); clear scope delineation (S. thermophilus vs. Lactococcus).

As the protospacer-spacer dynamics is indeed a key aspect of CRISPR-based defense systems, particularly in S. thermophilus, where CRISPR-Cas systems are highly prevalent, the authors can include heatmap (Annex 1) as a supplementary figure to demonstrate their analytical effort, and may briefly note or discuss the difficulties to draw definitive conclusions.

Reviewer #1 (Remarks to the Author):

In this manuscript Leprince et al. significantly extend our understanding of antiphage defenses in *S. thermophilus*, an important lactic acid bacterium of the dairy and fermentation industries. They show that although *St* is dominated by CR and RM systems that are well known, a deep bioinformatic dive into the genomes of this genus reveal other recently identified and obscure antiphage defense systems. Interestingly, they show that plasmid copy number and possibly plasmid replication can influence the efficacy of certain antiphage defense systems. This is an important finding as many in the field rely on ectopic expression of defense systems, often in heterologous hosts, to test phage defense. This argues that not only should native hosts be used, but the choice of plasmid should be considered. Finally, they show that they can achieve synergy with CR systems and other antiphage defenses that are more potent for phage protection than the individual systems alone. This is a well written manuscript that was easy to follow.

R: We thank the reviewer for their positive feedback on our manuscript and their constructive comments. Below, we provide detailed responses to the specific concerns raised.

1. The term “Resistome” is used throughout. This term is typically associated with antibiotic resistance. Perhaps using phage-resistome would help avoid any confusion.

R: We replaced the occurrences of “resistome” by “defensome” throughout the text as suggested by another reviewer.

2. Can the authors comment on the rare antiphage defense systems found in the *St* genomes (such as Theoris, Lamassu, etc.). Are such systems also uncommon in other streptococci, the Firmicutes as a whole, or is this a feature of *St*? Why would these systems be less prevalent compared to others?

R: While RM and CRISPR-Cas systems are widespread in bacteria, present in 83% and 39% of bacterial genomes, respectively, many defense systems are present in fewer than 20% of genomes or even in less than 1% (Tesson et al., 2022; Nat Commun). However, the prevalence of defense systems varies across bacterial species and phyla. For instance, the Wadjet system is more frequent in *Actinobacteria* (42%) compared to other phyla (7%) (Georjon et al., 2023; Microbiology). In *S. thermophilus*, defense systems that are extremely rare (<1% of strains) are also uncommon (<10%) in other streptococci, lactic acid bacteria (*Lactococcus* and *Lactobacillus*), *Bacillota* phylum, and bacteria in general (Table S4). However, Lamassu, which is present in only 0.4% of *S. thermophilus* strains, is found in 12.6% of bacterial genomes overall. Different bacterial taxa rely on distinct defense strategies, and it has been suggested that the varying prevalence of defense systems across taxa is primarily driven by phage diversity (Wu et al., 2024; Cell Host Microbe). In *S. thermophilus*, CRISPR-Cas systems appear to have been selected as the primary defense systems, as they are universally (100%) present in this species, far exceeding its overall bacterial prevalence (38%). A comparison of the defense system prevalence in *S. thermophilus* with related genera (*Streptococcus*, *Lactococcus*, and *Lactobacillus*) and the *Bacillota* phylum is presented in Table S4, using results available on the DefenseFinder website (<https://defensefinder.mdmlab.fr/wiki/refseq>). A sentence referring to this table was added to the text.

3. Can the authors comment on how they determined that no known anti-RM proteins were found in their phages? What anti-RM proteins were queried and does this include more recent anti-RM proteins that have been identified?

R: As described in the Materials and Methods section titled “ACR and methyltransferase predictions in phage genomes”, the tool dbAPIS was used to predict known counter-defense genes in phage genomes, including ACR and anti-RM genes. The dbAPIS database (Yan et al., 2024; Nucleic Acids Res) encompasses 14 types of anti-RM proteins, including Ocr, ArdA, and SieA, as well as recently identified proteins, including those reported by Silas et al. (2023). A relevance threshold was set at an e-value of $1e-30$, as prior experience with this tool and manual curation revealed numerous false positives (e.g., ACR hits matching capsid or large terminase genes). Additionally, these results were recently verified following the release of Anti-DefenseFinder (Tesson et al., 2025; Nucleic Acids Res), which, apart from ACRs, did not identify any other anti-defense genes in *S. thermophilus* phages.

4. In fig 4A when screening for recombinants that are phage resistant, how do the authors confirm that resistance is not driven by selection for cell wall modification/receptor mutants despite successful identification of homologous recombination of the system?

R: We did not specifically screen for receptor modifications. When isolating bacteriophage resistant mutants (BIMs) from *S. thermophilus*, they are primarily “CRISPR BIM” that have acquired a new spacer in either the CR1 or CR3 locus. Using PCR screening, we confirmed that the recombinant carrying a newly integrated defense system did not acquire new additional spacers. After assessing the efficiency of strains with a chromosomally integrated defense system, we also knocked out the defense system, which fully restored phage sensitivity (Fig. 5C and E, Fig. S4B-C). This confirmed that the observed resistance was solely attributable to the integrated defense system and not to other mutations that may have been selected during the phage selection process. This is described in the Materials and Methods section titled “Chromosomal insertion of defense systems”.

5. It appears that in the combination experiments only CR systems were paired with non-CR systems? What about synergy between RM and non-RM systems, or more uncommon systems with each other? Is synergy exclusive to CR or more widespread.

R: CRISPR-Cas systems are a core defense mechanism in *S. thermophilus*, with CR1, a type II-A CRISPR-Cas system, present in 100% of the strains. BIMs isolated after phage challenges predominantly acquire new spacers, most often in the CR1 locus, which enhances phage resistance. As a result, the dairy industry relies on these CRISPR-derived BIMs to adapt industrial strains to problematic virulent phages. This approach is widely used because BIMs are easily and naturally obtained, and the acquisition of additional spacers does not compromise the industrial properties of the strains.

The defense systems investigated in this study are not intended to replace CRISPR-Cas for generating phage-resistant *S. thermophilus* strains but rather to complement and reinforce CRISPR BIMs. Since any additional defense system introduced into *S. thermophilus* will coexist with at least one CRISPR-Cas system, this motivated our combination experiments. We previously demonstrated that CRISPR-Cas and RM systems are compatible and, when combined, provide enhanced phage resistance in *S. thermophilus* (Dupuis et al., 2013; Nat Commun). As such, we focused on accessory defense systems, which are the main subject

of this study. We specifically investigated the combined effects of Dodola, Thoeris, and Gabija, each of which conferred strong protection individually when cloned on pTRKL2. To assess their potential synergy, we evaluated their combined efficacy in killing assays using one defense system in the chromosome (Dodola or Gabija) and a second defense system provided *in trans* on pTRKL2. As the chromosomal system alone conferred high levels of protection, even at high MOIs, it was challenging to detect synergistic interactions. Nevertheless, additive effects were observed for Dodola-Gabija and Dodola-Thoeris combinations, while Gabija-Thoeris exhibited antagonistic effects at high MOI (Fig. S7).

Note that, to evaluate synergistic effects between non-CRISPR defense systems, we had to adapt our protocol (bacterial cultures were diluted 10 times for the assay, see Materials and Methods section “Killing curve”) to test higher MOIs (up to 50), as chromosomally integrated defense systems alone already conferred full protection at lower MOIs, thereby preventing the assessment of possible synergies. All experiments involving killing curves were repeated with this adapted protocol for consistency.

6. In terms of synergy, could the authors provide more insight into how some of these synergies might be happening mechanistically? I am not proposing additional experiments, just some comments on how they view this phenomenon happening.

R: Previous studies have shown that the *tmn* defense system synergizes with Gabija, Septu, and PrrC by co-opting their ATPase domains (Wu et al., 2024; Cell Host Microbe). In contrast, the synergy observed here with CRISPR-Cas (type II-A) involves three distinct defense systems with diverse functional domains (Fig. 4, Fig. S5-S6, and Table S5). While Gabija and Hachiman have a common helicase domain, Thoeris lacks this feature, suggesting that different molecular mechanisms mediate their respective interactions with CRISPR-Cas. Another case of synergy was reported in *Listeria*, where a type VI CRISPR-Cas system, which induces abortive infection through nonspecific RNA degradation, was shown to act cooperatively with a type I RM system (Williams et al., 2023; Nat Microbiol). The RM system cleaves the phage genome, thereby removing the source of phage transcripts and enabling recovery from cell dormancy, which is not observed in the absence of the RM system. Gabija, Hachiman, and Thoeris are also known to trigger abortive infection, either through nonspecific DNA degradation (Gabija and Hachiman) or NAD⁺ depletion (Thoeris). This raises the possibility that type II CRISPR-Cas systems may similarly mitigate the abortive infection phenotype, but it remains highly hypothetical at that stage. The paragraph discussing synergy has been improved to mention that the mechanisms underlying synergy were not investigated. Based on our experiments, we can only conclude that the combinations reduce the number of escape phages, which is of high importance in industrial settings.

Reviewer #2 (Remarks to the Author):

Summary

Leprince et al. explore the phage resistance mechanisms in *Streptococcus thermophilus* by examining the bacterium's defense systems beyond the well-characterized CRISPR-Cas and restriction-modification (RM) systems. Using bioinformatics tools (PADLOC and DefenseFinder), the authors analyzed the genomes of 263 *S. thermophilus* strains and identified 28 defense systems. Experimental validation of 17 systems against 14 phages revealed varying levels of resistance. The study also demonstrates that combining certain defense systems can provide synergistic protection, particularly against phages encoding anti-CRISPR proteins, with potential applications for developing more robust industrial *S. thermophilus* strains. Overall, the manuscript is presented very clearly, with a focused story and nice figures, but the study does not constitute a significant advancement to the field in its current state.

Review

The manuscript spends considerable time cataloguing the defense systems present in *Streptococcus thermophilus* strains using PADLOC and DefenseFinder. While a summary of the systems in these strains is presented clearly, much of this content is descriptive rather than transformative. The generation of such a dataset is trivial using these tools, and both resources already provide pre-computed databases of the defense systems in hundreds of *S. thermophilus* strains. The identification of 21 defense systems, although not explicitly explored in *S. thermophilus* previously, only includes systems that are already characterized in other bacteria and adds little theoretical or mechanistic understanding to the broader defense system field. In contrast, the initial development of DefenseFinder (published in Nature Communications >2 years ago) provided a significant advancement towards understanding the distribution of defense systems in all bacteria, not just a specific species.

The screening of systems in industrially relevant strains of *S. thermophilus* and *L. cremoris* is a refreshing detour from the usual workhorses of defense system research (*E. coli* and *B. subtilis*). Expanding the known defense profile of these systems beyond coliphages is valuable to the field. However, the authors do not advance our understanding of how these systems work. **Comparing these streptococcal and lactococcal phages against their coliphage counterparts may provide some insight into what is triggering these systems, but the authors do not explore this.**

R: We are currently attempting to isolate escape phages for these defense systems; however, this has proven to be challenging. It is now recognized that isolating escape phages can be particularly difficult for certain systems, such as defense systems (e.g., CapRel) which involves two distinct activation triggers (Zhang et al., 2024, Nature). A meaningful comparison between streptococcal phages and their coliphage counterparts would require extensive experimental work, likely constituting a dedicated study. For instance, in the study by Stokar-Avihail et al. (2023, Cell), the authors attempted to isolate escape mutants for 54 different defense systems and succeeded in only 15 cases, highlighting the complexity of such investigations.

The demonstration of activity for chromosomally encoded defenses is also important to the wider field of defense system research, as they are often only tested in the context of multicopy plasmids.

If the authors were able to demonstrate the efficacy of more types of defense systems in a chromosomal context, this would enhance the significance of the study for a broader audience.

R: In addition to PD-Lambda-1 and Dodola, we have now added the activity of three additional defense systems (Retron, Gabija, and Thoeris) in a chromosomal context, bringing the total number of individually tested defense systems to five. We also integrated a defense island composed of two systems (Hachiman and RM type II), into the genome of *S. thermophilus* DGCC7710. Two distinct integration strategies were used:

- Native-locus integration: PD-Lambda-1 and the Hachiman + RM II defense island were PCR-amplified from donor strains and introduced into DGCC7710 via natural transformation, allowing integration at their native chromosomal loci.
- Targeted-locus integration: Retron, Gabija, and Thoeris were inserted at the PD-Lambda-1 locus. These systems could not be tested in their native genomic contexts in DGCC7710, as they are part of larger genomic islands and the defense systems flanking genes are absent in this strain (see Fig. 5D). Integrating them at their native loci would have required the insertion of additional co-localized genes and removal of native DGCC7710 genes (including potentially uncharacterized defense elements) thereby compromising the ability to compare their individual activities.

Of note, we successfully integrated the AbiE system into *S. thermophilus* strain DGCC7891, but we were unable to generate a corresponding knockout strain. As a result, we did not include this strain in the study. Attempts to integrate the Kiwa system into DGCC7891 were unsuccessful. Additionally, no defense integrations were performed in strain DGCC8234, despite observed activity of certain systems in this background, as this strain cannot be transformed by natural competence. Activity of the integrated defense systems can be found in Fig. 5, Supplementary Fig. S4 (detailed titer data), and Fig. S9 (comparison of plasmid- vs chromosome-encoded system in liquid at different initial MOI).

Note that the integration of PD-Lambda-1 was repeated to match the strategy used here for the defense island, which was also integrated at its native chromosomal locus. In both cases, the recombination template was obtained directly by PCR amplification from the donor strain. This approach contrasts with our initial method, which involved designing a recombinant DNA fragment composed of the defense system from the donor and the flanking regions of the insertion site from the recipient strain. This newly integrated PD-Lambda-1 recombinant exhibited an efficiency comparable to the plasmid-encoded version. As a result, we revised our initial statement suggesting that chromosomally encoded defense systems may exhibit improved activity compared to their plasmid counterparts. Instead, based on the evaluation of other chromosomally integrated defense systems in both solid (Fig. 5 and S4B-C) and liquid (Fig. S9) assays, we observed that differences in efficiency between chromosomal and plasmid-encoded systems varied depending on the specific defense system and phage pair tested. This variability has important implications for conclusions drawn from experiments using plasmid-encoded defense systems, particularly for Abi phenotype assays (i.e., liquid assays at high MOI), which are increasingly used during defense system characterization.

The authors discuss a “pyramiding strategy,” wherein multiple defense systems are stacked in a particular strain to effectively prevent phage escape. While the idea is generally held that there is negative selection for defense systems in the absence of phage infection pressure because of a

fitness cost to maintaining these systems, some bacteria naturally harbor 50 (or more) different systems at a time. The present study would benefit from an experimental validation of the fitness costs of stacking defenses in this specific context. By assessing the impact on key industrial traits, the authors could provide a more practical assessment of the viability of this approach, adding relevance of their findings for industrial applications (which appears to be the goal for this work).

R: Thank you for your comment. We decided to assess the fitness cost of adding defense systems integrated into their native chromosomal location (See Fig. 5B-C), by monitoring bacterial growth and milk acidification. We evaluated the cost of adding one defense system (PD-Lambda-1) and one defense island composed of two defense systems (Hachiman and RM II) into the chromosome of our model strain DGCC7710. We showed that introducing either the individual defense system or the entire defense island had no impact on strain fitness, as bacterial growth and milk acidification rates remained unaffected.

We also evaluated the bacterial growth of other defense systems (Retron, Dodola, Thoeris, and Gabija) integrated into the chromosome but not at their native location. Our results showed that although the bacterial growth was not impacted for two out of the four tested (See Fig. 5E), milk coagulation abilities were affected for all of them and, for Dodola, the milk acidification rate was impaired (See Fig. S10), which could be explained by the fact that these defense systems were not integrated into their native location.

The main thesis of the study, that phage resistance is strengthened by the co-expression of complementary defense systems is already a well-established phenomenon. Where our understanding is lacking is why and how certain combinations of systems provide a synergistic effect, which the current work does not explore. Understanding the biology behind these observations will be crucial for effectively developing enhanced industrial strains. The observation that CRISPR-Cas + another defense is better than another defense alone in the presence of an anti-CRISPR is particularly interesting, but there is no evidence as to why this is happening? Does the addition of another defense system prevent the production of the anti-CRISPR, allowing normal levels of CRISPR-Cas protection? Is there physical interaction or signaling between CRISPR-Cas and the additional system? Is the expression of multiple toxic components increasing the general stress response of the host, limiting infection?

R: To also answer another reviewer's comment, we modified our protocol for the killing assays to allow testing at higher MOIs. Specifically, bacterial cultures adjusted to an OD₆₀₀ of 0.2 were diluted 10-fold, enabling us to test MOIs up to 50, which also resulted in bacteria being tested in earlier exponential growth phase. Using this protocol, we repeated all the combination experiments, testing MOIs ranging from 0.0005 to 50. Synergistic effects were observed not only with the ACR-containing phage D5691, but also with phage 2972, which does not encode an ACR (Fig. 4B-C). These synergies were consistent across all combinations of CRISPR-Cas with Gabija, Hachiman, or Thoeris and observed at higher MOIs. While exploring the molecular mechanisms underlying these synergistic effects is indeed of interest, we believe that such investigations are beyond the scope of the current study.

Overall, the manuscript is presented very clearly, with a consistent and easy to follow throughline and high-quality figures. The inclusion of raw tabular data in the supplementary material makes the bioinformatics portion of the work easily verifiable. The identification of defenses and anti-

defenses in *S. thermophilus* and their phages is interesting from an industrial perspective but lacks in the theoretical or mechanistic understanding that it brings to the wider field. Further exploration into the mechanisms of *S. thermophilus*-encoded systems, the mechanisms of synergy between systems and/or assessment of the real-world viability of engineering industrial strains through the addition of defenses, would greatly enhance the significance of this paper. In its current state, it may be better suited to a more-specialized journal.

R: In addition to the previously mentioned adaptations, such as the assessment of fitness costs and the chromosomal integration of three additional defense systems as well as a defense island, we also evaluated the efficacy of PD-Lambda-1 and the defense island (Hachiman + RM II) in controlling phage infection under milk fermentation conditions, which represent the natural environment of *S. thermophilus*. Furthermore, we added a paragraph to the Discussion to emphasize aspects likely to appeal to a broader audience which are addressed in this study, specifically the importance of evaluating defense systems within their native host and genomic context to better reflect “real-life” conditions.

Additional points:

L20 & throughout: “Resistome” is a vague term that is often used to describe antibiotic resistance rather than phage defense, consider re-phrasing to “phage resistome”, “defensome” or something similar.

R: We replaced the occurrences of “resistome” by “defensome” throughout the text.

Fig1C: The scale for branch length is missing a description of the unit being represented (nucleotide substitutions per site?).

R: A description of the scale associated with the phylogenetic tree (amino acid substitutions per site) has been added to the legend.

L188: In the statement “This co-occurrence suggests a complementary role, potentially targeting different phages and enhancing the global defense of a given strain”, it is unclear what is meant by “This co-occurrence”. Initially it appears to refer to a co-occurrence between AbiD and SoFic, but it seems these systems are mutually exclusive in *S. thermophilus*. If it is instead saying that multiple homologues of these systems being present in the same strain suggests that they are targeting different phages, then this is rather speculative when we don’t know how these systems work, and the similarity of these homologues has not been compared.

R: This is indeed a hypothesis that co-occurring homologues of defense systems could target different phages, which we showed is the case for the tested homologues of Gao19 for example. However, we agree that we did not specifically investigate co-occurring homologues, and the limited mechanistic insight into these systems (AbiD, AbiE, AbiH, and SoFIC) makes this hypothesis highly speculative. Therefore, we have removed the sentence from the text.

L216: The copy number of pNZ123 is incredibly high, far beyond what is standard in the field for testing the efficacy of defense systems. Considering that most of the systems identified in *S. thermophilus* were chromosomally encoded (suggesting a natural low copy number) and that many defense systems are toxic when overexpressed, why was this vector considered in the first place?

R: The vector pNZ123 is widely used in *S. thermophilus*, notably for the characterization of ACR encoded by phages infecting this bacterium (e.g., Hynes et al., 2017; Nat Microbiol and Hynes et al., 2018; Nat Commun). This use initially motivated its consideration in our study. We compared it with other vectors less commonly used, such as pTRKL2 and pTRKH2, which have lower copy numbers. While defense systems tested in pNZ123 were less efficient when cloned into lower copy number plasmids, *S. thermophilus* strains transformed with pNZ123 carrying defense systems did not exhibit growth defects typically associated with the overexpression of these systems. Ultimately, we opted to use pTRKL2 over pTRKH2 because of its lower copy number, despite lower transformation efficiencies compared to pNZ123 and pTRKH2.

L222: It is not clear that “all tested defense systems” provided at least a 1-log reduction in plaquing, for example PD-lambda-1 CIRM956 and RosmerTA CNRZ1066 did not.

R: We acknowledge that the original phrasing was unclear. Here, we meant that all 18 distinct defense systems tested conferred resistance to at least one phage. This statement does not refer to the different homologues, which are addressed in the following paragraph. The sentence has been revised for clarity.

L226: Phage replication may have been reduced below the limit of detection in these specific experimental conditions, but it is not necessarily “abolished”.

R: The portion of the sentence containing the term “abolished” was removed, keeping only the part stating that certain systems provide up to 7-log protection.

L227: The apparent broad-spectrum defense provided by some defense systems can often be attributed to the toxicity of these systems when expressed in a heterologous host rather than anti-phage activity per-se. Do the authors see any growth defects in the strains transformed with these systems?

R: No growth defect was observed when the different strains were transformed with the plasmid encoded defense systems tested in this study (See Fig. S9 for growth curves of selected defense systems either plasmid- or chromosomally-encoded).

L248: It is unclear whether defense system homologues are exhibiting different defense profiles due to inherent differences in the defense systems themselves or different compatibilities with the host strain, can the authors comment on this?

R: The tested homologues originate from different strains, each with distinct sets of defense systems. Similarly, the testing strains possess different defense system profiles compared to the original strains from which the homologues were derived. Given the small number of strains analyzed and the likelihood that several defense systems remain uncharacterized, it is currently not possible to draw conclusions regarding potential incompatibilities between the defense systems being evaluated and those present in the strains used for the assessment.

L287: Why were only these two systems tested? Were more tested but not effective?

R: Initially, only two defense systems were tested, as the primary objective was to provide proof of concept that these systems could be integrated into the chromosome of an industrial strain while conferring phage resistance. To answer your concern, we integrated

three additional defense systems in the chromosome as well as a defense island composed of two defense systems (See comment above).

L226: See the previous point regarding the claim that defense is “abolished”.

R: The sentence was modified to: “Combining CR1 immunity with either Gabija, Hachiman, or Thoeris reduced phage replication below the limit of detection, with no escape phages observed in spot tests”.

L307 & throughout: “X was transformed into Y” is incorrect phrasing, it should be “Y was transformed with X”.

R: This was adapted accordingly.

L308: “Has a one spacer”?

R: This typo has been corrected.

FigS3: Typically, plaque forming units per mL are reported for these types of phage assays (sometimes in addition to the efficiency of plaquing data shown here), and the test for significance is between the difference in plaque forming units for the control (no defense) and the strain with defense. While change in EOP is useful for visualization, it can also obfuscate some of the underlying data that may change the interpretation of the results, so PFU should also be shown. The type of t-test performed here also requires an adjustment for multiple comparisons if each defense system is being compared to the same control. While a reduction in plaquing is often obvious without the statistics, it is important that statistics are presented clearly and correctly if they are to be included. Error bars should also be shown (as in FigS4A).

R: The data shown in Fig. S3 correspond to the heatmap presented in Fig. 2, which summarizes the efficiency of each defense system against the tested phages. Such heatmaps commonly represent EOP-based data (e.g., fold protection, fold reduction in EOP, or log-EOP) to enable straightforward comparison across multiple defense systems. Given that statistical analyses for this heatmap were conducted on EOP values, we believe it is more relevant to present the EOP rather than titres. A defense system was considered protective if the EOP was statistically different from 1 and showed at least a 1-log reduction. The specific statistical tests used are now clarified in the Materials and Methods section under “Assessment of defense system efficiency” and “Killing curve assay” and multiple testing correction was performed using the Benjamini–Hochberg method.

In addition to Fig. S3, EOP data are also presented in two main figures: Fig. 3D and Fig. 5E. In Fig. 5, which compares plasmid- and chromosome-encoded defense systems, we chose to display EOP rather than phage titers to avoid overcrowding the figure. Showing titers would have required including two different controls for each phage (DGCC7710 WT and DGCC7710 pTRKL2), resulting in five bars per phage and making the figure overly complex. However, to ensure transparency, we have included the corresponding titer data in Fig. S4. To maintain consistency across the manuscript, Fig. 3D which compare the activity of Gao19 alleles, also presents EOP data, with the underlying titer values likewise provided in Fig. S4. For Gao19 alleles, initial spot tests for strains DGCC7710, DGCC8234, DGCC8849, and MG1363 were conducted on different days, each with its own empty vector control. To enable direct comparison between the ST3 and ST109 alleles, we repeated these assays using a shared empty vector control. As a result, we present only three bars per

phage in the final figure: one for the control, one for allele ST3, and one for allele ST109. We also corrected the text accordingly, especially as we noticed that we originally inverted ST3 and ST109 allele results.

FigS3: The distinction between countable plaques and 'fuzzy' zones of lysis is a nice addition to the data, but it's a little hard to differentiate between the filled/hollow circles, could these be represented in a clearer way, perhaps different colors?

R: The hollow circles indicating lysis zones were changed to red hollow circles in Figures 3D, 4D, and 5B-D, and Supplementary Figures.

L373: Population collapse at high MOI is often attributed to many systems said to have an "abortive infection" response. However, this collapse could also be explained by phage-related damage or overcoming defense at high MOIs, rather than activity of the system itself. This is unfortunately almost never differentiated, but speculation about the mechanism and evolution of such systems should be made carefully (see Afraim & Eldar, 2023; <https://doi.org/10.1016/j.tim.2023.05.002>).

R: We agree that the Abi phenotype and mechanism should be distinguished, which is why we specifically refer to the Abi phenotype in Table 1. The reference mentioned is already cited in the discussion section on Abi systems (Reference 54). We have also clarified our hypothesis by specifying that an Abi mechanism has not been established for most of the accessory defense systems identified in *S. thermophilus*.

L482: e^{-30} is confusing (incorrect?) notation, consider changing to 1×10^{-30} or $1e^{-30}$ if this is what is meant.

R: This was changed to $1e^{-30}$.

L567 & throughout: Explanation of the statistical tests performed are lacking. I.e. what were the tests comparing? Were corrections for multiple testing applied where appropriate?

R: This was clarified in the text (see also above comment).

Reviewer #3 (Remarks to the Author):

In this manuscript, the authors make a significant contribution to our understanding of anti-phage defense systems in *S. thermophilus*. They identify novel defense systems and assess their anti-phage activity in vivo, incorporating a broader range of *S. thermophilus* strains. This research is particularly valuable for those studying microbial defense mechanisms. The authors also describe an intriguing phenomenon: certain defense systems show enhanced efficiency when integrated into the chromosome or expressed via low-copy plasmids. Additionally, they demonstrate the synergistic effects of combining some of these defense systems with the CRISPR system. Overall, this study not only deepens our understanding of anti-phage defenses in *S. thermophilus* but also opens up new possibilities for developing more robust industrial strains.

R: We thank the reviewer for their appreciation of the manuscript, helpful remarks and comments. Below, we provide detailed responses to each of the specific remarks.

I have a few specific comments:

1. The finding that defense systems are more efficient when integrated into the chromosome or expressed from low-copy plasmids is particularly interesting. Could the authors provide more explanation on this? Are these differences in efficiency related to variations in protein expression levels?

R: In *S. thermophilus*, plasmids are rare and most defense systems are chromosomally encoded in a single copy. Thus, using low-copy-number plasmids closely mimic the native genomic context of these systems. Moreover, the enhanced resistance observed for certain systems may reflect chromosomal integration alongside their native promoters. In addition to Dodola and PD-Lambda-1, we integrated three additional individual systems (Gabija, Retron, and Thoeris) into the chromosome of DGCC7710 and assessed their activity on solid and in liquid media (Fig. 5E and Fig. S4B-C). While the level of protection varied depending on the system and phage tested, we found that some systems, such as Retron, were more efficient when plasmid-encoded, whereas others, like Gabija, generally performed better when chromosomally integrated. In most cases, however, no clear difference in activity was observed between the two contexts.

Notably, the integration of PD-Lambda-1 was repeated to be consistent with the strategy used for integrating a defense island at its native chromosomal locus (i.e., recombination template obtained through PCR amplification of the donor strain; see Fig. 5B-C and Materials and Methods section "Chromosomal insertion of defense systems"). This newly chromosomally integrated PD-Lambda-1 derivative exhibited an efficiency comparable to the plasmid-encoded version. As a result, we revised our initial statement suggesting that chromosomally encoded defense systems may systematically exhibit improved activity compared to their plasmid counterparts. Instead, based on the evaluation of multiple chromosomally integrated systems, we conclude that the impact of genomic context on defense system efficiency is system-dependent, with no consistent trend toward higher or lower activity on plasmid versus chromosome. However, the differences observed underscore the importance of evaluating and characterizing defense systems in their native host and genomic context whenever possible. This variability has implications for conclusions drawn from experiments using plasmid-encoded defense systems, particularly for Abi phenotype assays (i.e., liquid assays at high MOI), which are increasingly used during defense system characterization (refer to discussion).

2. While the synergistic effects between the CRISPR system and other anti-phage defenses were explored, the study hints at the potential for developing more resilient industrial strains of *S. thermophilus*. Given the broad-spectrum activity observed in the Gabija, Dodola, and Thoeris systems, it would be interesting to determine whether these systems work synergistically or antagonistically in combination.

CRISPR-Cas systems are ubiquitous in *S. thermophilus*, with CR1, a type II-A CRISPR-Cas system, serving as a core defense mechanism present in 100% of strains. Phage-resistant mutants isolated after phage challenges predominantly acquire new spacers, most often in the CR1 locus, which enhances phage resistance. As a result, the dairy industry relies on these CRISPR-derived BIMs to adapt industrial strains to problematic phages. This approach is widely used because BIMs are easily obtained, and the acquisition of additional spacers does not compromise the industrial properties of the strains. The defense systems investigated in this study are not intended to replace CRISPR-Cas for generating phage-resistant *S. thermophilus* strains but rather to complement and reinforce CRISPR BIMs. Since any additional defense system introduced into *S. thermophilus* must coexist with at least one CRISPR-Cas system, this motivated our combination experiments.

Here, we further investigated the combined effects of non-CRISPR defense systems as Dodola, Thoeris, and Gabija, each of which conferred strong protection individually when tested on pTRKL2. To assess their potential synergy, we evaluated their combined efficacy in killing assays using one defense system in the chromosome (Dodola or Gabija) and a second defense system provided *in trans* on pTRKL2. As the chromosomal system alone conferred high levels of protection, even at high MOIs, it was challenging to detect synergistic interactions. Nevertheless, additive effects were observed for Dodola-Gabija and Dodola-Thoeris combinations, while Gabija-Thoeris exhibited antagonistic effects at high MOI. These results are provided in Fig. S7.

Note that, to evaluate synergistic effects between non-CRISPR defense systems, we had to adapt our protocol (bacterial cultures were diluted 10 times for the assay, see Materials and Methods section “Killing curve”) to test higher MOIs (up to 50), as chromosomally integrated defense systems alone already conferred full protection at lower MOIs, thereby preventing the assessment of possible synergies. All experiments involving killing curves were repeated with this adapted protocol for consistency.

3. To enhance reader comprehension, it would be helpful to include a detailed description of the genetic composition and functional domains of these defense systems. This would provide clearer insights into how they function.

R: As recommended by another reviewer, Figure 3C has been revised to illustrate the genetic composition and functional domains of the defense systems evaluated in this study. Additionally, detailed information about the functional domains identified by InterPro is now provided in Table S5.

4. Lastly, there is a minor typographical error on line 253: "SIR2-Her system" should be corrected to "SIR2-HerA."

R: Thank you for bringing this to our attention. The error has been corrected.

Reviewer #4 (Remarks to the Author):

Comments to the authors:

This manuscript provides a comprehensive analysis of the genomes of 263 *Streptococcus thermophilus* strains, identifying 28 distinct phage-resistant defense systems. These defense systems were detected using two computational tools: DefenseFinder and PADLOC. Furthermore, 17 systems and several combined systems were also experimentally validated. The methodology sounds good. This significantly expands our understanding of the phage-resistance mechanisms in this species. The study offers a valuable resource, giving a broad perspective on the diverse array of defense systems present. The complementary nature of these defense systems suggests the potential to develop more robust industrial strains, which would be of great interest to the dairy industry. Overall, this work represents an important contribution to the field.

R: We thank the reviewer for their appreciation of the manuscript, helpful remarks, and comments. Below, we provide detailed responses to each remark.

Major concerns:

--The composition or genetic architectures of the defence systems reported in this study should be drawn for better illustration and understanding. See the Figure 2D in the reference "Diverse enzymatic activities mediate antiviral immunity in prokaryotes. Science. 2020 Aug 28;369(6507):1077-1084" and Table 1 of "Systematic discovery of antiphage defence systems in the microbial pangenome. Science. 2018 Mar 2;359(6379):eaar4120."

R: As recommended, Figure 3C has been revised to illustrate the genetic composition and functional domains of the defense systems evaluated in this study. Additionally, detailed information about the functional domains identified by InterPro is now provided in Table S5.

--The protospacer sequences in phage genomes and the corresponding spacer sequences in bacterial CRISPR arrays also represent the evolutionary "arms race" between phages and their bacterial hosts. In the Results section "Streptococcal phages have strategies to counter CRISPR-Cas and RM defenses", the authors should further investigate the patterns and dynamics of these protospacer-spacer interactions to provide more insights into the dynamic co-evolutionary processes shaping the phage-host relationship.

As an example, using the CRISPRimmunity tool, we analyzed the *Streptococcus thermophilus* strain ST3 and found 2 type II-A CRISPR-Cas systems with a total of 40 spacers matching protospacers on 216 *Streptococcus* phage genomes. The detailed information can be accessed via the following link:

http://www.microbiome-bigdata.com/CRISPRimmunity/index/datas_crisprimmunity/complete_strain/GCF_002286255.1_ASM228625v1

R: The protospacer-spacer dynamics is indeed a key aspect of the co-evolutionary arms race between phages and bacteria, particularly in *S. thermophilus*, where CRISPR-Cas systems are highly prevalent. To conduct an analysis of these dynamics, we extracted spacers from the CR1 locus (i.e., the most active in spacer acquisition) across the 263 *S. thermophilus* strains analyzed in this study. These spacers were then compared to a

custom database of the 191 phage genomes in our dataset using BLASTn. We provide a heatmap (Annex 1, see below) illustrating the number of spacer matches per strain corresponding to protospacers in each phage genome, with phages ordered according to their phylogeny. The two P738 phages are not included, as none of the analyzed strains contained matching spacers for these genomes.

While some strains (e.g., GCF_025187525 and GCF_903891245.1) show a higher number of spacer matches (particularly against *Moineavirus* and *Brussowvirus* phage genera), no consistent trend emerges from this analysis. This may be due to differences in phage abundance and genetic diversity across phage genera. Moreover, since both the bacterial and phage genomes were obtained from public database, detailed information about host-phage relationships is often missing, which limits our ability to draw definitive conclusions. By analysing subset of strains (example in Annex 2, see below), we also did not observe a clear inverse correlation between high CRISPR spacer content and reduced presence of non-CRISPR defense systems. Although it could be hypothesized that strains with more active CRISPR-Cas systems may rely less on other defense mechanisms, our preliminary analysis do not support this trend.

Future investigations into protospacer-spacer dynamics in *S. thermophilus* should expand beyond spacer matching to include analysis of PAM sequences and integrate host-related factors such as *eps* and *rgp* genes (encoding potential phage receptors) and phage receptor-binding protein sequences. Combining these with known phage-host associations and literature data could provide deeper insights into host range and co-evolutionary dynamics. However, such a comprehensive analysis lies beyond the scope of the current study, and we have chosen not to include our preliminary analysis to the current work.

--Although the study demonstrated the benefits of combining multiple defence systems, it did not look in depth at the potential costs of such 'pyramid' stacking of defences, such as the reduction in bacterial growth and fermentation performance. In addition, the performance of the pyramiding strategy, where multiple defense systems are stacked within the same strain, and the mixing strategy or cocktail of strains of individual defense systems, need to be further compared and weighed.

R: The fitness cost of the integration an additional defense system (PD-Lambda-1) or a defense island composed of two defense systems (Hachiman and RM II) into their native chromosomal location was evaluated by assessing the impact on bacterial growth and milk acidification rate (Fig. 5B-C). In addition, the impact on bacterial growth of four other defense systems integrated in the chromosome but not in their native location (as they are part of larger genomic islands which would involve exchanging the island of the recipient strain with the one of interest and therefore potentially deleting uncharacterized defense system) was also evaluated through bacterial growth monitoring (Fig. 5D). The pyramiding and mixing strategies were compared using killing assay with three defenses systems combination (See Fig. 4F). This confirmed the observation of Gandon et al. (2024; Proc Biol Sci) that the pyramiding strategy provides better resistance than the mixing strategy.

Minor concerns:

--The study mainly focused on the phage resistance defence system of *Streptococcus thermophilus* (*Streptococcus lactis*) but did not address other important dairy fermenting bacteria such as *Lactococcus lactis*. This limits the general applicability of the results.

R: *S. thermophilus* and dairy lactococcal species use distinct defense strategies against phages. In *S. thermophilus*, CRISPR-Cas systems are ubiquitous, whereas in lactococcal species, they are found in only a limited number of strains (Millen et al., 2012; PLoS One). Moreover, while most defense systems in *S. thermophilus* are chromosomally encoded, dairy-associated *Lactococcus* species harbor numerous plasmids (Ainsworth et al., 2014; FEMS Microbiol Rev), many of which carry defense systems (Grafakou et al., 2024; Nucleic Acids Res). Thus, plasmid conjugation is a common strategy in the dairy industry to generate phage-resistant *Lactococcus* strains, a mechanism that is not available for *S. thermophilus*. Consequently, findings from *S. thermophilus* cannot necessarily be extrapolated to *Lactococcus*, and vice versa. Other groups have focused on characterizing the *Lactococcus* defensome (Grafakou et al., 2024; Nucleic Acids Res). Table S4 presents a comparison of the prevalence of *S. thermophilus* defense systems in *Lactococcus* and *Lactobacillus*, two genera relevant to the dairy industry.

--Although the study systematically analysed defense systems on the genomes of 263 *S. thermophilus* strains using DefenseFinder and PADLOC and identified 28 known types of defence systems, it is likely that there are still some novel defence systems that have not been identified. Further exploration may still be needed.

R: We acknowledge the strong likelihood of additional, uncharacterized defense systems present in *S. thermophilus*. However, a thorough investigation and analysis of these potential new defense systems fall beyond the scope of this study.

Overall, this is an excellent piece of work that has important implications for a deeper understanding of the coevolutionary relationship between bacteria and phages and for the development of more robust industrial strains. But there is still a need to further expand the scope of the study, validate the performance of the combined defence systems for industrial applications, and evaluate the potential costs of multiple defence systems in order to achieve a more comprehensive and reliable solution.

R: We hope to have answered above those concerns.

Annex 1 – Analysis of spacer–protospacer dynamics in the CR1 loci of *S. thermophilus* strains. Spacers were extracted from the CR1 loci of the 263 *S. thermophilus* strains and compared to a custom database of 191 phage genomes using BLASTn. The phylogenetic tree on the left illustrates the relationships between the bacterial strains, with branch lengths representing the number of amino acid substitutions per site. Phages are shown along the x-axis. The heatmap displays the number of spacers per strain with a 100% match to protospacers in the corresponding phages, as indicated by the color scale. The bar plot above the heatmap indicates the total number of matching spacers per phage (range: 0–450), while the bar plot to the left shows the total number of matching spacers per strain (range: 0–880).

Annex 2 – Analysis of spacer–protospacer dynamics in the CR1 loci of selected *S. thermophilus* strains. Eight *S. thermophilus* strains were selected for in-depth analysis. The heatmap shows the number of CR1 spacers per strain that perfectly match (100% identity) protospacers in the phage genomes, with values indicated by the color scale. The presence of CRISPR-Cas loci (types 1–4) and restriction-modification (RM) systems (types I–IV) in each strain is represented by colored dots. Additionally, the number of other known defense systems, as well as the total number of defense systems per strain, is indicated.

Reviewer #4 (Remarks to the Author):

The authors have fully addressed major/minor comments through adding new figures/tables (Fig 3C, Table S4-5, Fig 4F, 5B-D); additional bioinformatic analysis (spacer-protospacer matching); clear scope delineation (*S. thermophilus* vs. *Lactococcus*).

As the protospacer-spacer dynamics is indeed a key aspect of CRISPR-based defense systems, particularly in *S. thermophilus*, where CRISPR-Cas systems are highly prevalent, the authors can include heatmap (Annex 1) as a supplementary figure to demonstrate their analytical effort and may briefly note or discuss the difficulties to draw definitive conclusions.

R: We thank the reviewer for his comment. Annex 1 was integrated as Supplementary Figure 3 and a short paragraph discussing this figure was added as follows:

L151: “One strategy used by phages to evade CRISPR-Cas targeting is through mutations in their protospacer sequences or PAM motifs. Consequently, protospacer–spacer dynamics represent a key aspect of the co-evolutionary arms race between phages and bacteria, particularly in *S. thermophilus*, where CRISPR-Cas systems are highly prevalent (Supplementary Figure 3).”

The method section was also adapted to include the methodology behind this analysis.